# SET LEARNING FOR ACCURATE AND CALIBRATED MODELS

Lukas Muttenthaler[1,2,3,*,†], Robert A. Vandermeulen[1,2,*], Qiuyi (Richard) Zhang[3], Thomas Unterthiner[3], and Klaus-Robert Müller[1,2,3,4,5]

[1]Machine Learning Group, Technische Universität Berlin, Germany
[2]Berlin Institute for the Foundations of Learning and Data, Berlin, Germany
[3]Google DeepMind
[4]Department of Artificial Intelligence, Korea University, Seoul
[5]Max Planck Institute for Informatics, Saarbrücken, Germany

## ABSTRACT

Model overconfidence and poor calibration are common in machine learning and difficult to account for when applying standard empirical risk minimization. In this work, we propose a novel method to alleviate these problems that we call odd-$k$-out learning (OKO), which minimizes the cross-entropy error for sets rather than for single examples. This naturally allows the model to capture correlations across data examples and achieves both better accuracy and calibration, especially in limited training data and class-imbalanced regimes. Perhaps surprisingly, OKO often yields better calibration even when training with hard labels and dropping any additional calibration parameter tuning, such as temperature scaling. We demonstrate this in extensive experimental analyses and provide a mathematical theory to interpret our findings. We emphasize that OKO is a general framework that can be easily adapted to many settings and a trained model can be applied to single examples at inference time, without significant run-time overhead or architecture changes.

## 1 INTRODUCTION

In machine learning, a classifier is typically trained to minimize cross-entropy on individual examples rather than on sets of examples. By construction, this paradigm ignores information that may be found in correlations between sets of data. Therefore, we present *odd-k-out* learning (OKO), a new training framework based on learning from sets. It draws inspiration from the *odd-one-out* task which is commonly used in the cognitive sciences to infer notions of object similarity from human decision-making processes (Robilotto & Zaidi, 2004; Fukuzawa et al., 1988; Hebart et al., 2020; Muttenthaler et al., 2022; 2023a). The odd-one-out task is a similarity task where subjects choose the most similar pair in a set of objects. We use an adapted version of that task to learn better model parameters while not making any changes to the architecture (see Fig. 1; a).

Standard classification training often yields overconfident classifiers that are not well-calibrated (Müller et al., 2019; Guo et al., 2017; Minderer et al., 2021). Classically, calibration has been treated as an orthogonal problem to accuracy. Miscalibration has been observed to severely worsen while accuracy improves, an interesting phenomenon attributed to over-parametrization, reduced regularization, and biased loss functions (Guo et al., 2017; Vaicenavicius et al., 2019; Roelofs et al., 2022). Even log-likelihood — a proper scoring rule — was accused of biasing network weights to better classification accuracy at the expense of well-calibrated probabilities (Guo et al., 2017; Roelofs et al., 2022). Other scoring rules were proposed that are differentiable versions of calibrative measures but these approximations can be crude (Karandikar et al., 2021). Thus, calibration methods are often treated as an afterthought, comprised of ad-hoc post-processing procedures that require an additional hold-out dataset and monotonically transform the output probabilities, usually without affecting the learned model parameters or accuracy.

---

[*]Equal contributions.
[†]Work partly done while a Student Researcher at Google DeepMind.

Calibration is inherently a performance metric on sets of data; so we propose training the classifier on sets of examples rather than individual samples to find models that yield accurate calibration without ad-hoc post-processing. This is especially crucial in low-data and class-imbalanced settings, for which there is surprisingly little work on calibration (Dal Pozzolo et al., 2015b).

Various techniques have been proposed to improve accuracy for imbalanced datasets (Branco et al., 2016; Johnson & Khoshgoftaar, 2019), which are typically based on non-uniform class sampling or reweighting of the loss function. However, neural nets can still easily overfit to the few training examples for the rare classes (Wang & Japkowicz, 2004). There is growing interest in the development of new techniques for handling class imbalance (Johnson & Khoshgoftaar, 2019; Iscen et al., 2021; Parisot et al., 2022; Guha Roy et al., 2022). Such techniques are adapted variants of non-uniform sampling, often focusing exclusively on accuracy, and ignoring model calibration. However, techniques for mitigating the effects of imbalance on classification accuracy do not improve calibration for minority instances and standard calibration procedures tend to systematically underestimate the probabilities for minority class instances (Wallace & Dahabreh, 2012). Moreover, it is widely known that direct undersampling of overrepresented classes modifies the training set distribution and introduces probabilistic biases (Dal Pozzolo et al., 2015a). Bayesian prior readjustments were introduced to manipulate posterior probabilities for ameliorating that issue (Dal Pozzolo et al., 2015b).

It is known that hard labels tend to induce extreme logit values and therefore cause overconfidence in model predictions (Hinton et al., 2015; Bellinger et al., 2020). *Label smoothing* has been proposed to improve model calibration by changing the cross-entropy targets rather than scaling the logits after training (Müller et al., 2019; Carratino et al., 2022). Label smoothing, in combination with batch balancing — uniformly sampling over the classes rather than uniformly sampling over all samples in the data (see Appx. B.3), achieves promising results on heavy-tail classification benchmarks, i.e. datasets that contain many classes with few samples and a few classes with many samples (Bellinger et al., 2020). Yet, all these methods ignore the need for accuracy on the underrepresented classes, generally lack rigorous theoretical grounding, and require fine-tuned parameters for good empirical performance, such as the noise parameter for label smoothing, or the scaling parameter for temperature scaling for which additional held-out data is required.

In contrast to the popular philosophy of training for accuracy and then calibrating, we pose our main question: Can we provide a training framework to learn network parameters that simultaneously obtain better accuracy and calibration, especially with class imbalance?

**Contributions.** *Indeed, we find that OKO achieves better calibration and uncertainty estimates than standard cross-entropy training. The benefits of OKO over vanilla cross-entropy are even more pronounced in limited training data settings and with heavy-tailed class distributions.* [1]

**Empirical. First**, through extensive experiments, we show that OKO often achieves *better accuracy* while being *better or equally well calibrated* than other methods for improving calibration, especially in low data regimes and for heavy-tailed class distribution settings (see Fig. 1; b). **Second**, OKO is a principled approach that changes the learning objective by presenting a model with *sets of examples* instead of individual examples, as calibration is inherently a metric on sets. As such, OKO does not introduce additional hyperparameters for post-training tuning or require careful warping of the label distribution via a noise parameter as in label smoothing (see Fig. 1). **Third**, surprisingly, this differently posed set learning problem results in *smoothed logits* that yield *accurate calibration*, although models are trained using hard labels. **Fourth**, we emphasize that OKO is extremely easy to plug into any model architecture, as it provides a general training framework that does not modify the model architecture and can therefore be applied to single examples at test time exactly like any network trained via single-example learning (see Fig. 1; a). The training complexity scales linearly in $O(|\mathcal{S}|)$ where $|\mathcal{S}|$ denotes the number of examples in a set and hence introduces *little computational overhead* during training. **Last**, in few-shot settings, OKO achieves compellingly *low calibration and classification errors* (see Fig. 1; b). Notably, OKO improves test accuracy for 10-shot MNIST by $8.59\%$ over the best previously reported results (Liu et al., 2022).

**Theoretical.** To get a better understanding of OKO's exceptional calibration performance, we offer mathematical analyses that show why OKO yields logit values that are not as strongly encouraged to diverge as in vanilla cross-entropy training. We develop a new scoring rule that measures excess confidence on a per-datapoint basis to provably demonstrate improved calibration. This scoring rule

---

[1] A `JAX` implementation of OKO is publicly available at: `https://github.com/LukasMut/OKO`

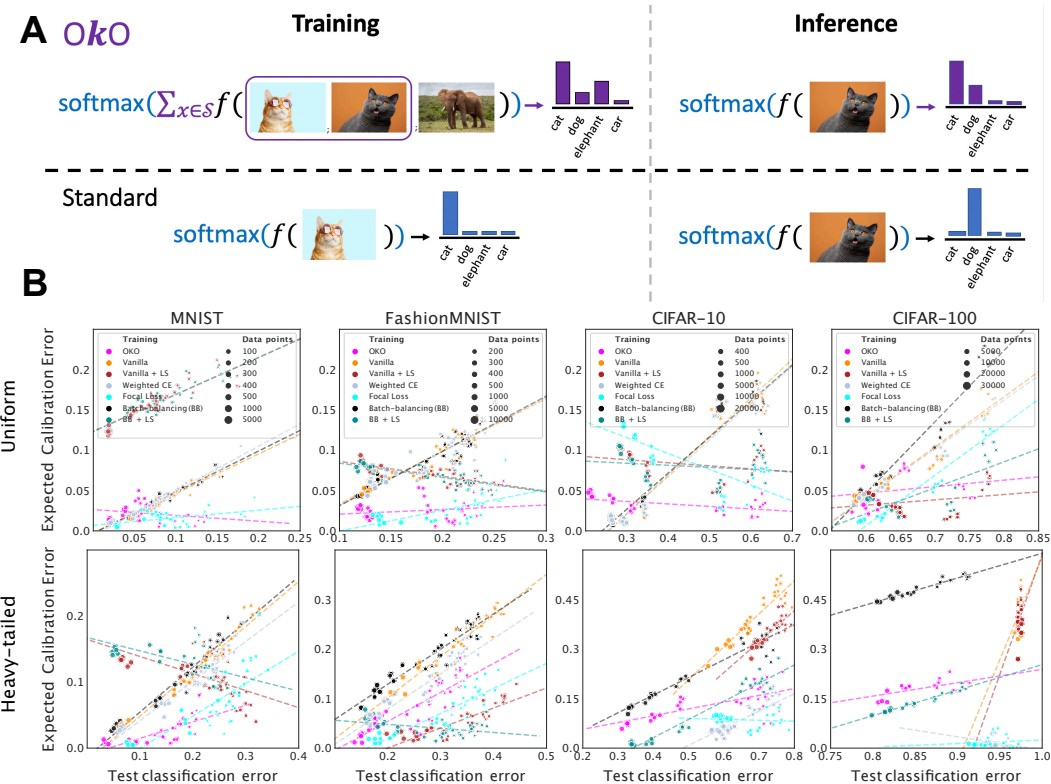

Figure 1: **A**: OKO minimizes cross-entropy on sets of examples rather than on single examples and naturally yields smoothed logits after training. At inference time it can be applied to single examples without additional computational overhead. **B**: Expected calibration error as a function of the classification error. Each point in the graph represents the performance of a single seed; there are five for every number of training data points. For each dataset, every model was evaluated on the same test set. Dashed diagonal lines indicate a linear regression fit. Top: Uniform class distribution during training. Bottom: Heavy-tailed class distribution during training.

compares the predictive entropies and cross-entropies, and for calibrated predictors, we show that our measure is consistent in that the average excess confidence is 0. By using this new scoring rule we demonstrate that OKO implicitly performs a form of *entropic regularization*, giving insight into how it prevents excess confidence in certain low entropy regions.

## 2 RELATED WORK

The *odd-one-out* task has been widely used in the cognitive sciences to infer notions of object similarity from human participants (Robilotto & Zaidi, 2004; Hebart et al., 2020; Muttenthaler et al., 2022; 2023a), and first uses are slowly percolating into machine learning: Fernando et al. (2017) trained a self-supervised video understanding network by predicting which one out of three sequences was in reverse time order, Locatello et al. (2020); Mohammadi et al. (2020) used comparisons between samples as weak supervision target. Muttenthaler et al. (2023b) use human odd-one-out choices to improve pretrained representations for few-shot learning and anomaly detection tasks. However, none of these works investigated calibration or provided any theory for (odd-one-out) set learning.

Improving calibration is of practical interest for many applications. However, deep neural networks often appear badly calibrated (Guo et al., 2017). Even though this depends on the concrete architecture used, scaling up a model usually increases accuracy at the cost of calibration (Minderer et al., 2021). Many post-hoc approaches to increase calibration have been proposed, such as temperature scaling (Platt et al., 1999), isotonic regression (Zadrozny & Elkan, 2002; Niculescu-Mizil & Caruana, 2005), and Bayesian binning (Naeini et al., 2015), while improving calibration during training is a less explored topic. Most related to our approach are techniques that use data augmentations that blend

different inputs together (Thulasidasan et al., 2019) or use ensembles to combine representations (Lakshminarayanan et al., 2017). However, none of these works examined calibration for sets of data. The task of classifying sets of instances is known as *multiple instance learning* (Carbonneau et al., 2018). It is desirable to leverage the set structure, instead of simply using a concatenated representation of the examples in each set. A common approach is to pool representations, either by mean pooling, which is akin to OKO, or max pooling (Feng & Zhou, 2017; Pinheiro & Collobert, 2014). Other approaches include the use of permutation invariant networks (Zaheer et al., 2017) or attention mechanisms (Ilse et al., 2018; Cordonnier et al., 2021). Also related are Error-correcting output codingDietterich & Bakiri (1995), which looks at multiclass classification by training several different binary classifiers. We are unaware of work that leverages set learning for improving the calibration of standard cross-entropy training.

Learning from imbalanced data has a long history in machine learning (Japkowicz & Stephen, 2002; He & Garcia, 2009; Branco et al., 2016). Approaches usually center around resampling the training data (Chawla et al., 2002; Drummond & Holte, 2003; Liu et al., 2009) or modifying the loss function (Chen et al., 2004; Ting, 2000; Wallace et al., 2011; Khan et al., 2018; Cui et al., 2019; Lin et al., 2020; Du et al., 2023), or combinations thereof (Huang et al., 2016; Liu et al., 2019; Tian et al., 2022). Transfer learning (Wang et al., 2017; Zhong et al., 2019; Parisot et al., 2022), self-supervised learning (Yang & Xu, 2020; Kang et al., 2021), or ensembles of experts (Collell et al., 2018; Wang et al., 2021; Guha Roy et al., 2022; Cui et al., 2023; Jiang et al., 2023) can also be helpful for rare classes. Our method is a novel way to improve performance on imbalanced data at excellent calibration.

## 3  METHOD

Here we present the core contribution of this work, *odd-$k$-out* training (OKO). In OKO a model is simultaneously presented with multiple data points. At least two of these data points are from the same class, while the remaining $k$ data points are each from a different class, i.e., the odd-$k$-outs, or *odd classes*. The objective is to predict the *pair class*. This forces a model to consider correlations between sets of examples that would otherwise be ignored in standard, single-example learning.

**Notation.** More formally, we are interested in the classification setting on a training set $\mathcal{D} = \{(x_1, y_1), \ldots, (x_n, y_n)\} \subset \mathbb{R}^d \times [C]$ of inputs $x_i$ and labels $y_i$ from $C$ classes. The number of odd classes $k$ is chosen such that $k + 1 \leq C$. We construct an OKO training example $\mathcal{S}$ as follows:

Let $\mathcal{X}_c$ be the set of all training inputs, $x_i$, such that $y_i = c$. One first, uniformly at random, selects a label $y' \in [C]$ and sets $y'_1 = y'_2 = y'$ as the *pair* class. Next $y'_3, \ldots, y'_{k+2}$ are sampled uniformly without replacement from $[C] \setminus \{y'\}$ as the *odd* classes. Finally $x'_1, \ldots, x'_{k+2}$ are selected uniformly at random from $\mathcal{X}_{y'_1}, \ldots, \mathcal{X}_{y'_{k+2}}$, while enforcing $x'_1 \neq x'_2$. So $x'_1$ and $x'_2$ have the same class label, $y'$, and $x'_3, \ldots, x'_{k+2}$ all have unique class labels not equal to $y'$. A training example is then $\mathcal{S} = \left((x'_1, y'_1), \ldots, (x'_{k+2}, y'_{k+2})\right)$. Let $\mathcal{S}_x := \left(x'_1, \ldots, x'_{k+2}\right)$ and $\mathcal{S}_y = \left(y'_1, \ldots, y'_{k+2}\right)$. Alg. 1 describes the sampling process. The distribution of $\mathcal{S}$ according to Alg. 1 is $\mathscr{A}$.

---

**Algorithm 1** $\mathscr{A}$ - OKO set sampling

---

**Input:** $\mathcal{D}, C, k$          $\triangleright$ $C$ is the number of classes and $k$ is the number of odd classes
**Output:** $\mathcal{S}_x, \mathcal{S}_y, y'$
    $y' \sim \mathcal{U}\left([C]\right)$          $\triangleright$ Sample a pair class for constructing the set
    $y'_1 \leftarrow y', y'_2 \leftarrow y'$
    $y'_3, \ldots, y'_{k+2} \overset{NR}{\sim} \mathcal{U}\left([C] \setminus \{y'\}\right)$      $\triangleright$ Sample $k$ odd classes *without replacement*
    **for** $i = 3, \ldots, k + 2$ **do**
       $x'_i \leftarrow \mathcal{U}\left(\mathcal{X}_{y'_i}\right)$      $\triangleright$ Choose a representative input for each of the $k + 2$ set members
    **end for**
    $\mathcal{S}_x \leftarrow \left(x'_1, \ldots, x'_{k+2}\right); \mathcal{S}_y \leftarrow \left(y'_1, \ldots, y'_{k+2}\right)$

---

**OKO objective** For a tuple of vectors, $\mathcal{S}_x := \left( x'_1, \ldots, x'_{k+2} \right)$ and a neural network function $f_\theta$ parameterized by $\theta$, we define $f_\theta(\mathcal{S}_x) := \sum_{i=1}^{k+2} f_\theta \left( x'_i \right)$ and the following *soft* loss for a fixed set $\mathcal{S}$:

$$\ell_{\text{oko}}^{\text{soft}} \left( \mathcal{S}_y, f_\theta \left( \mathcal{S}_x \right) \right) := -((k+2)^{-1} \sum_{i=1}^{k+2} \boldsymbol{e}_{y'_i})^\top \log \left[ \text{softmax} \left( f_\theta \left( \mathcal{S}_x \right) \right) \right], \tag{1}$$

where $\boldsymbol{e}_a \in \mathbb{R}^C$ is the indicator vector at index $a$ and $\text{softmax}$ denotes the softmax function. The soft loss encourages a model to learn the distribution of all labels in the set $\mathcal{S}$. One may also consider the case where the network is trained to identify the most common class $y'$, yielding the *hard* loss:

$$\ell_{\text{oko}}^{\text{hard}} \left( \mathcal{S}_y, f_\theta \left( \mathcal{S}_x \right) \right) := -\boldsymbol{e}_{y'}^\top \log \left[ \text{softmax} \left( f_\theta(\mathcal{S}_x) \right) \right]. \tag{2}$$

In ablation experiments, we found the hard loss to always outperform the soft loss and have thus chosen not to include experimental results for the soft loss in the main text (see Appx. F.5 for further details). For OKO set sampling, $\mathcal{S} = (\mathcal{S}_x, \mathcal{S}_y) \sim \mathscr{A}$, the empirical risk is $\mathbb{E}_{\mathcal{S} \sim \mathscr{A}} \left[ \ell_{\text{oko}}^{\text{soft}} \left( \mathcal{S}_y, f_\theta \left( \mathcal{S}_x \right) \right) \right]$.

## 4 PROPERTIES OF OKO

Here we theoretically analyze aspects of the OKO loss that are relevant to calibration. First, via rigorous analysis of a simple problem setting, we demonstrate that OKO implicitly performs regularization by preventing models from overfitting to regions with few samples, thereby lowering certainty for predictions in those regions. We refer to these regions as *low-entropy* regions, where, for all inputs $x$ in such a region, $p(y|x)$ has most of its probability mass assigned to one class. Second, we introduce and analyze a novel measure for calibration that is based on the model output entropy rather than label entropy. This has the advantage of directly examining the cross-entropy error as a function of the model uncertainties and evaluate its correspondence. Additionally, in Appx. C, we include an analysis of a simplified loss landscape of OKO, and show that it less strongly encourages logit outputs to diverge compared to vanilla cross-entropy, while allowing more flexibility than label smoothing.

**OKO is less certain in low-data regions.** Imagine a dataset where the majority of the data is noisy. Specifically, most of the data share the same feature vector but have different class labels — *one-feature-to-many-classes*, and each class has $0 < \epsilon \ll 1$ fraction of data points in a low-entropy region in which the data points are clustered together by one class label — *many-features-to-one-class*.

In such a high-noise dataset it is likely that the low-entropy regions are mislabeled. If $f_\theta$ has high capacity and was fitted via vanilla regression, it would overfit to low-entropy regions by classifying them with high certainty since those examples are well-separated from the noise. As mentioned in the previous section, even label smoothing only slightly alleviates overfitting (Müller et al., 2019).

A simple example should illustrate the previously mentioned setting: We will demonstrate that, for this example, the OKO method assigns low certainty to low-entropy regions. To this end we will consider a binary classification problem on an input space consisting of three elements. Let $\mathbb{F}$ be the set of all functions in $\{0, 1, 2\} \mapsto \mathbb{R}^2$, this is analogous to the space of all possible models that can be used to classify, e.g., $f_\theta$ from before.

Now let $\mathscr{A}_\epsilon$ with $\epsilon \in [0, 1]$ be defined as in Alg. 1 where the proportion of the training data, $(x_1, y_1), \ldots, (x_n, y_n)$, having specific values is defined in Table 1. Note that $n$ does not matter for the results that we present here. For $0 < \epsilon \ll 1$ this indicates that the vast majority of the data has $x = 0$ with equal probability of both labels. The remainder of the data is split between $x = 1$ and $x = 2$ where the label always matches the feature and is thus a zero-entropy region. For this setting, we introduce the following theorem which we proof in Appx. D.

|  | $y_i = 1$ | $y_i = 2$ |
|---|---|---|
| $x_i = 0$ | $(1 - \epsilon)/2$ | $(1 - \epsilon)/2$ |
| $x_i = 1$ | $\epsilon/2$ | $0$ |
| $x_i = 2$ | $0$ | $\epsilon/2$ |

Table 1: PMF for Theorem 1

**Theorem 1.** *For all $\epsilon \in (0, 1)$ there exists $f_\epsilon$ such that*

$$f_\epsilon \in \arg \min_{f \in \mathbb{F}} \mathbb{E}_{\mathcal{S} \sim \mathscr{A}_\epsilon} \left[ \ell_{\text{oko}}^{\text{hard}} \left( \mathcal{S}_y, f_\theta \left( \mathcal{S}_x \right) \right) \right]. \tag{3}$$

*Furthermore, for any collection of such minimizers indexed by $\epsilon$, $f_\epsilon$, as $\epsilon \to 0$, then* $\text{softmax} \left( f_\epsilon(0) \right) \to [1/2, 1/2]$, $\text{softmax} \left( f_\epsilon(1) \right) \to [2/3, 1/3]$, *and* $\text{softmax} \left( f_\epsilon(2) \right) \to [1/3, 2/3]$.

The key from Theorem 1 is that, although $x = 1$ or $x = 2$ have zero entropy and are low-entropy regions, OKO is still uncertain about these points because they occur infrequently. This may be interpreted as the network manifesting epistemic uncertainty (label uncertainty in an input region due to having few training samples) as aleatoric uncertainty (uncertainty in an input region due to the intrinsic variance in the labels for that region) in the OKO test time outputs. The desirability of this property may be somewhat dependent on the application. While it is conceivable that such regions do indeed have low aleatoric uncertainty, and that the few samples do indeed characterize the entire region, for safety-critical applications it is often desirable for the network to err towards uncertainty.

**Relative Cross-Entropy for Calibration.** We introduce an entropy-based measure of sample calibration and demonstrate empirically that it is a useful measure of calibration, along with theoretical justification for its utility. In a sense, our measure is inspired by the log-likelihood scoring function and is a normalized scoring rule that gives sample-wise probabilistic insight (for details see Appx. E).

**Definition 1.** *Let the relative cross-entropy of distributions $P, Q$ be $RC(P, Q) = H(P, Q) - H(Q)$.*

Since $RC$ can be computed for each $(y, \hat{y})$ datapoint, it is a scoring rule. The relative cross-entropy is very similar to KL divergence but with a different entropy term. However, unlike the KL divergence, it is not always non-negative. In fact, note that if an incorrect prediction is overconfident, then $RC(y, \hat{y}) \to \infty$ is extremely positive, implying that $RC$ captures some measure of excess confidence. Specifically, we can show when the predictions are inaccurate, we have a provable deviation.

**Lemma 1.** *For hard labels $y$, if $\boldsymbol{e}_y^\top \hat{y} \leq 1/|C|$, then $RC(y, \hat{y}) \geq 0$.*

Furthermore, we show that $RC$ captures some notion of calibration when averaged across all data points. Specifically, when a predictor is perfectly calibrated, its average $RC$, a measure of excess confidence, should be 0. Note that $RC$ is no longer proper due to this zero mean.

**Lemma 2.** *If $\hat{y}$ is a predictor that is perfectly calibrated across $\mathcal{D}$, then the average excess confidence, as measured by relative cross-entropy, is $\mathbb{E}_{(x,y)\sim\mathcal{D}}[RC(y, \hat{y}(x))] = 0$*

## 5 EXPERIMENTAL RESULTS

In this section, we present experimental results for both generalization performance and model calibration. In general, model calibration and generalization performance are orthogonal quantities. A classifier can show strong generalization performance while being poorly calibrated, and, vice versa, a classifier can be well-calibrated although its generalization performance is weak. Here, we are equally interested in both quantities.

**Experimental details.** For every experiment we present in this section, we use a simple randomly-initialized CNN for MNIST and FashionMNIST and ResNet18 and ResNet34 architectures (He et al., 2016) for CIFAR-10 and CIFAR-100 respectively. We use standard SGD with momentum and schedule the learning rate via cosine annealing. We select hyperparameters and train every model until convergence on a held-out validation set. To examine generalization performance and model calibration in low data regimes, we vary the number of training data points while holding the number of test data points fixed. We report accuracy for the official test sets of MNIST, FashionMNIST, CIFAR-10, and CIFAR-100. We are specifically interested in heavy-tailed class distributions. Since heavy-tailed class distributions are a special rather than a standard classification setting, we report experimental results for both uniform and heavy-tailed class distributions during training. We consider heavy-tailed class distributions with probability mass $p = 0.9$ distributed uniformly across three overrepresented classes and $(1 - p) = 0.1$ distributed across the remaining 7 or 97 underrepresented classes respectively. In ablation experiments we have seen that, although odd class examples are crucial, OKO is not sensitive to the particular choice of $k$ (see App. F.4). Therefore, we set $k$ in OKO to 1 for all experiments. Note that $k = 1$ results in the computationally least expensive version of OKO. Since preliminary experiments have shown that generalization performance can be boosted by predicting the odd class using an additional classification head, in the following we report results for a version of OKO with $k = 1$ where in addition to the pair class prediction (see Eq. 2) a model is trained to classify the odd class with a second classification head that is discarded at inference time.

For simplicity and fairness of comparing against single example methods, we set the maximum number of randomly sampled sets to the total number of training data points $n_{\text{train}}$ in every setting. This is guaranteed to yield the same number of gradient updates as standard cross-entropy training.

**Training methods.** Alongside OKO, we consider six different baseline methods for comparing generalization performance and seven different methods for investigating model calibration: 1.) Standard maximum-likelihood estimation (see Eq. 4 in Appx. B.1), 2.) Vanilla + label smoothing (LS; Müller et al., 2019), 3.) Focal Loss (Lin et al., 2017), 4.) Cross-entropy error reweighting (see Eq. 5 in Appx. B.2), 5.) Batch-balancing (BB; see Alg. 2 in Appx. B.3), 6.) BB + LS, 7.) BB + temperature scaling (TS; $\tau = 2.0$). We consider label smoothing because it yields significantly better calibration than using hard labels for training neural nets and equivalent model calibration to temperature scaling (Müller et al., 2019). We deliberately ignore temperature scaling for generalization performance analyses because it does not change the $\arg\max$ of a classifier's predicted probability distribution after training and therefore yields the same test accuracy as BB.

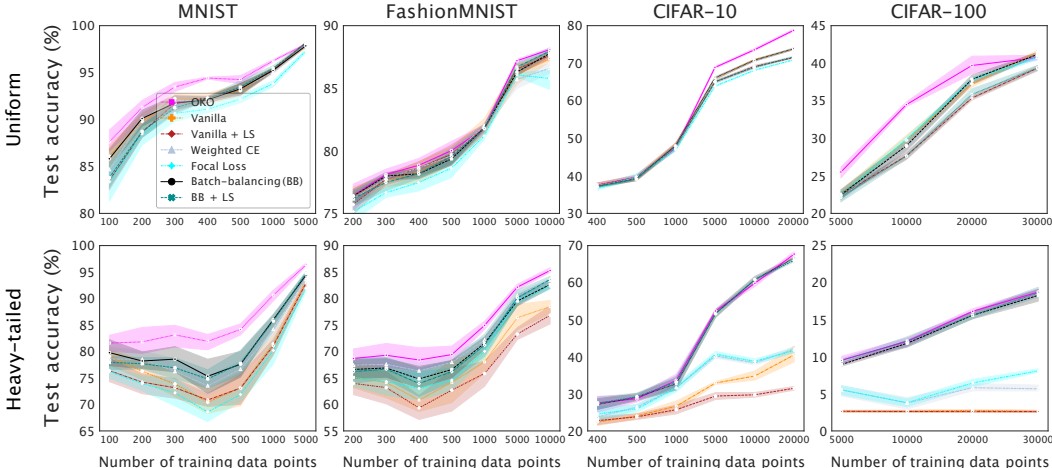

Figure 2: Test set accuracy in % as a function of different numbers of data points used during training. Error bands depict 95% CIs and are computed over five random seeds for all training settings and methods. Top: Uniform class distribution during training. Bottom: Heavy-tailed class distribution.

**Generalization.** For both uniform and heavy-tailed class distribution settings, OKO either outperforms or performs on par with the best baseline approaches considered in our analyses across all four datasets (see Fig. 2). We observe the most substantial improvements over the baseline approaches for both balanced and heavy-tailed MNIST, heavy-tailed FashionMNIST, and balanced CIFAR-10 and CIFAR-100. For 10-shot MNIST OKO achieves an average test set accuracy of $87.62\%$, with the best random seed achieving $90.14\%$. This improves upon the previously reported best accuracy by $8.59\%$ (Liu et al., 2022). For 20-shot and 50-shot MNIST, OKO improves upon the previously reported best test set accuracies by $2.85\%$ and $1.81\%$ respectively (Liu et al., 2022). OKO achieves the strongest average generalization performance across all datasets and class distribution settings (see Tab. 2). Improvements over the other training methods are most substantial for heavy-tailed training settings.

Table 2: Test set accuracy averaged across all training settings shown in Fig. 2.

| Training \ Distribution | MNIST | | FashionMNIST | | CIFAR-10 | | CIFAR-100 | |
|---|---|---|---|---|---|---|---|---|
| | uniform | heavy-tailed | uniform | heavy-tailed | uniform | heavy-tailed | uniform | heavy-tailed |
| Vanilla | 92.34% | 78.21% | 81.05% | 68.16% | 55.84% | 30.24% | 32.72% | 02.68% |
| Vanilla + LS | 91.84% | 77.38% | 81.10% | 66.46% | 55.06% | 27.16% | 31.20% | 02.62% |
| Weighted CE | 92.14% | 80.18% | 79.52% | 71.45% | 55.74% | 33.76% | 32.42% | 05.20% |
| Focal Loss (Lin et al., 2017) | 90.98% | 76.42% | 80.13% | 69.86% | 54.39% | 33.89% | 32.72% | 05.98% |
| BB | 92.31% | 81.42% | 81.11% | 71.24% | 55.87% | 44.69% | 32.63% | 13.67% |
| BB + LS | 91.86% | 80.81% | 81.12% | 71.13% | 54.96% | 44.72% | 31.26% | 13.96% |
| OKO (ours) | **93.62%** | **85.67%** | **81.49%** | **74.02%** | **57.63%** | **44.95%** | **35.11%** | **14.13%** |

**Calibration.** We present different qualitative and quantitative results for model calibration. Although model calibration is an orthogonal quantity to generalization performance, it is equally important for the deployment of machine learning models.

**Reliability.** The reliability of a model can be measured by looking at a model's accuracy as a function of its confidence. An optimally calibrated classifier is a model whose predicted class is

correct with probability $\hat{p}_\theta(x)$, where $\hat{p}_\theta(x)$ is the confidence of a model's prediction, i.e., optimal calibration occurs along the diagonal of a reliability diagram (see Fig. 3). OKO's reliability lies along the diagonal substantially more often than to any competing method. This is quantified by lower Expected Calibration Errors (see Fig. 1; 10) of OKO compared to the other methods. Its calibration is on par with BB + LS or BB + TS in some settings. In Fig. 3, we show reliability diagrams for MNIST, FashionMNIST, CIFAR-10, and CIFAR-100 averaged over all training settings using a uniform class distribution. Reliability diagrams for the heavy-tail training settings can be found in Appx. F.

**Uncertainty.** Entropy is a measure of uncertainty and therefore can be used to quantify the confidence of a classifier's prediction. Here, we examine the distribution of entropies of the predicted probability distributions for individual test data points as a function of (in-)correct predictions.

An optimally calibrated classifier has much density at entropy close to $\log 1$ and little density at entropy close to $\log C$ for correct predictions, and, vice versa, small density at entropy close to $\log 1$ and much density at entropy close to $\log C$ for incorrect predictions, irrespective of whether classes were in the tail or the mode of the training class distribution. In Fig. 4, we show the distribution of entropies of the models' probabilistic outputs partitioned into correct and incorrect predictions respectively for MNIST and FashionMNIST across all training settings with heavy-tailed class distributions. We observe that label smoothing does alleviate the overconfidence problem to some extent, but is worse calibrated than OKO. More entropy visualizations can be found in Appx. F.

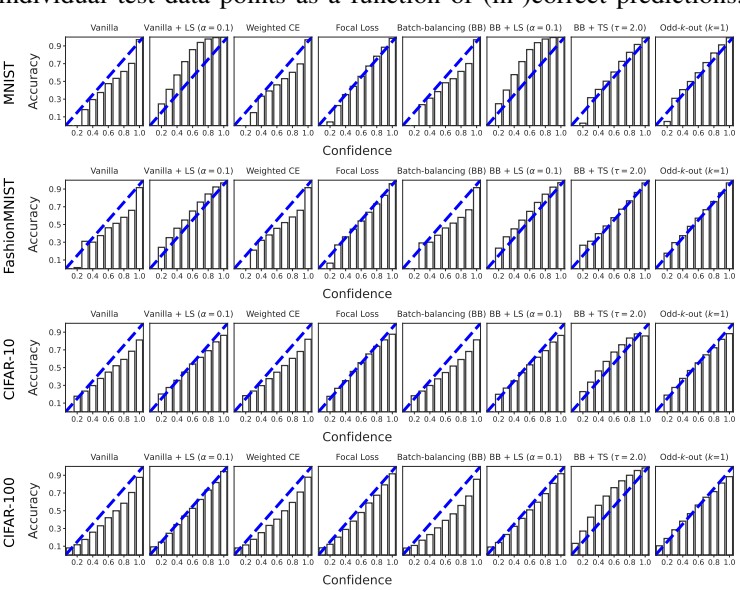

Figure 3: Reliability diagrams for balanced datasets. Confidence and accuracy scores were averaged over random seeds and the number of training data points. Dashed diagonal lines indicate perfect calibration.

**ECE.** ECE is a widely used scoring rule to measure a classifier's calibration. It is complementary to reliability diagrams (see Fig. 3) in that it quantifies the reliability of a model's confidence with a single score, whereas reliability diagrams qualitatively demonstrate model calibration. A high ECE indicates poor calibration, whereas a classifier that achieves a low ECE is generally well-calibrated. Aside from CIFAR-100 where batch-balancing in combination label smoothing shows slightly lower ECEs than OKO, OKO achieves lower ECE scores than any other method across training settings (see Fig. 1 in §1 and Fig 10 in Appx. F).

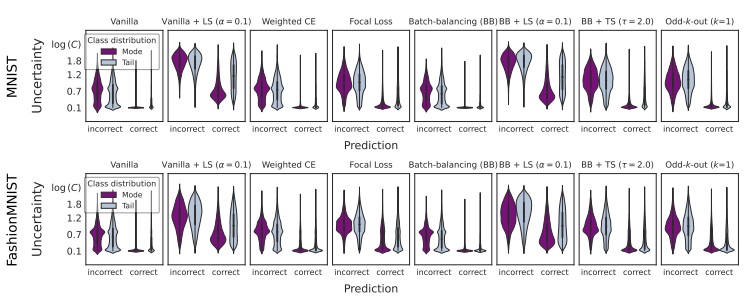

Figure 4: Here, we show the distribution of entropies of the predicted probability distributions for individual test data points across all heavy-tailed training settings partitioned into correct and incorrect predictions respectively.

*RC.* Here, we demonstrate empirically that our novel entropy-based measure of datapoint calibration is a useful measure of calibration. Following Def. 1 and Lemma 2 in §4, we quantify the average excess confidence $RC\left(y, \hat{y}(x)\right)$ by measuring the mean absolute difference (MAE) between $\bar{H}(P, Q)$ and $\bar{H}(Q)$ for the different number of training data point settings (see Fig. 5). We find that OKO

Table 3: MAE between entropies and cross-entropies averaged over the entire test set for different numbers of training data points. Lower is better and therefore bolded.

| Training \ Distribution | MNIST | | FashionMNIST | | CIFAR-10 | | CIFAR-100 | |
|---|---|---|---|---|---|---|---|---|
| | uniform | heavy-tailed | uniform | heavy-tailed | uniform | heavy-tailed | uniform | heavy-tailed |
| Vanilla | 0.189 | 0.723 | 0.455 | 1.075 | 0.330 | 1.845 | 0.708 | 2.638 |
| Vanilla + LS | 0.475 | 0.342 | 0.243 | 0.119 | 0.230 | 1.158 | 0.236 | 2.171 |
| Weighted CE | 0.207 | 0.558 | 0.505 | 0.758 | 0.315 | 0.366 | 0.705 | **0.189** |
| Focal Loss (Lin et al., 2017) | **0.044** | 0.333 | 0.107 | 0.308 | 0.222 | **0.296** | 0.526 | 0.198 |
| BB | 0.201 | 0.709 | 0.455 | 1.275 | 0.330 | 1.438 | 0.918 | 6.362 |
| BB + LS | 0.475 | 0.380 | 0.240 | **0.114** | 0.225 | 0.471 | 0.337 | 1.141 |
| OKO (ours) | 0.073 | **0.094** | **0.080** | 0.334 | **0.116** | 0.498 | **0.314** | 1.164 |

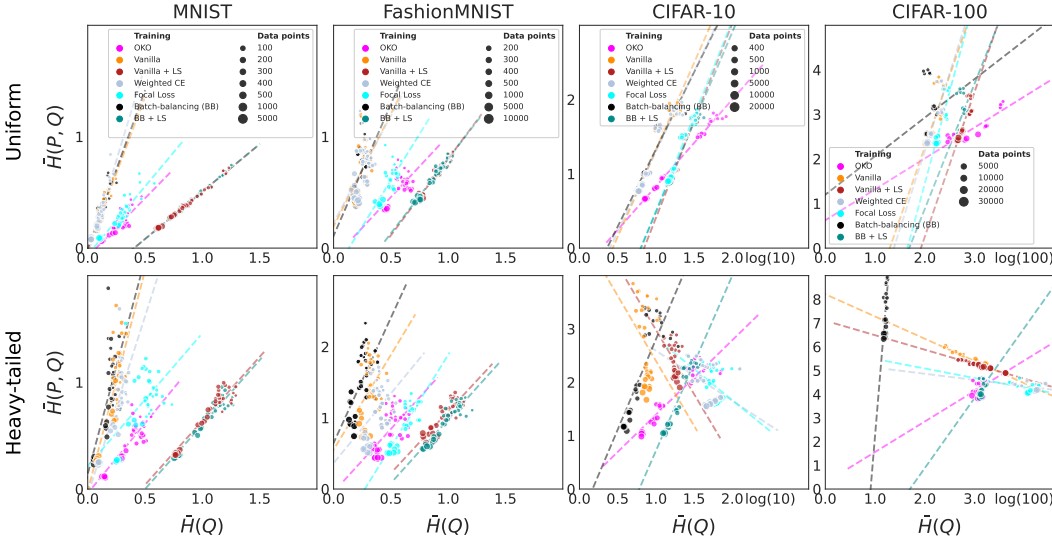

Figure 5: For different numbers of training data points, OKO achieves a substantially lower MAE for the average cross-entropy error between true and predicted class distributions and the average entropy of the predictions — across both uniform and heavy-tailed class distributions during training.

achieves the lowest MAE for all balanced training settings and is among the top-2 or top-3 training methods with the lowest MAE for the heavy-tailed training settings (see Tab. 3).

## 6 CONCLUSION

In standard empirical risk minimization, a classifier minimizes the risk on individual examples; thereby ignoring more complex correlations that may emerge when considering sets of data. Our proposed odd-$k$-out (OKO) framework addresses this caveat — inspired by the *odd-one-out* task used in the cognitive sciences (Hebart et al., 2020; Muttenthaler et al., 2022; 2023a). Specifically, in OKO, a classifier learns from sets of data, leveraging the odd-one-out task rather than single example classification (see Fig. 1). We find that OKO yields well-calibrated model predictions, being better or equally well-calibrated as models that are either trained with label smoothing or whose logits are scaled with a temperature parameter found via grid search after training (see §5). This alleviates the ubiquitous calibration problem in ML in a more principled manner. In addition to being well-calibrated, OKO achieves better test set accuracy than all training approaches considered in our analyses (see Tab. 2). Improvements are particularly pronounced for the heavy-tailed class distribution settings.

OKO modifies the training objective into a classification problem for sets of data. We provide theoretical analyses that demonstrate why OKO yields smoother logits than standard cross-entropy, as corroborated by empirical results. OKO does not require any grid search over an additional hyperparameter. While OKO is trained on sets, at test time it can be applied to single examples exactly like any model trained via a standard single example loss. The training complexity scales linearly in $O(|\mathcal{S}|)$ where $|\mathcal{S}|$ denotes the number of examples in a set and hence introduces little computational overhead during training.

One caveat of OKO is that classes are treated as semantically equally distant — similar to standard cross-entropy training. An objective function that better reflects global similarity structure may alleviate this limitation. In addition, we remark that we have developed OKO only for supervised learning with labeled data. It may thus be interesting to extend OKO to self-supervised learning.

We expect OKO to benefit areas that are in need of reliable aleatoric uncertainty estimates but suffer from a lack of training data — such as medicine, physics, or chemistry, where data collection is costly and class distributions are often heavy-tailed.

## ACKNOWLEDGMENTS

LM, RV, and KRM acknowledge funding from the German Federal Ministry of Education and Research (BMBF) for the grants BIFOLD22B and BIFOLD23B. LM acknowledges support through the Google Research Collabs Programme. We thank Rodolphe Jenatton for helpful comments on an earlier version of the manuscript.

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

# A   OKO CLASSIFICATION HEAD

Here, we demonstrate how easily OKO can be applied to any neural network model in practice, irrespective of its architecture. OKO does not require an additional set of parameters. OKO essentially is just a sum over the logits in a set of inputs. Below we provide JAX code for the classification head that is used for OKO. During training, logits obtained from the classification head are summed across the inputs in a set. At inference time, the classification head is applied to single inputs just as any standard classification head.

```python
import flax.linen as nn
import jax.numpy as jnp
from einops import rearrange
from jax import vmap

Array = jnp.ndarray

class OKOHead(nn.Module):
    num_classes: int # number of classes in the data
    k: int # number of odd classes in a set

    def setup(self) -> None:
        self.clf = nn.Dense(self.num_classes)
        self.scard = self.k+2 # set card is number of odd classes + 2

    def set_sum(self, x: Array) -> Array:
        """Aggregate the logits across all examples in a set."""
        x = rearrange(x, "(b scard) d -> b scard d", scard=self.scard)
        dots = vmap(self.clf, in_axes=1, out_axes=1)(x)
        set_logits = dots.sum(axis=1) # set sum
        return set_logits

    @nn.compact
    def __call__(self, x: Array, train: bool) -> Array:
        # x \in \mathbb{R}^{b(k+2) \times d}
        if train:
            logits = self.set_sum(x_p)
        else:
            logits = self.clf(x)
        return logits
```

Listing 1: OKO classification head implemented in JAX.

# B   BACKGROUND

## B.1   PROBLEM SETTING

In the heavy-tailed class distribution setting, we are interested in the classification setting where a classifier has access to training data $\mathcal{D} = \{(x_1, y_1), \ldots, (x_n, y_n)\} \subset \mathbb{R}^d \times [C]$ consisting of inputs $x_i$ and labels $y_i$ from $C$ classes. For $c \in [C]$, $\mathcal{D}_c \subset \mathcal{D}$ will denote those samples with label $c$, so $\mathcal{D} = \bigcup_{c=1}^{C} \mathcal{D}_c$. Class imbalance occurs when $|\mathcal{D}_c| \gg |\mathcal{D}_{c'}|$ for some $c$ and $c'$. Let $e_a \in \mathbb{R}^C$ be the indicator vector at index $a$ and softmax be the softmax function. Let $\mu_n$ be the uniform empirical measure of $\mathcal{D}$.

**Cross-entropy**. The cross-entropy error between the original and the predicted labels is the following risk function when averaged over all $n$ data points in the dataset $\mathcal{D}$,

$$\mathcal{L}_{\text{vanilla}}(\mathcal{D}, \theta) := \mathbb{E}_{(X,Y) \sim \mu_n} [\ell_{\text{x-ent.}}(e_Y, \text{softmax}(f_\theta(X)))] = -\frac{1}{n} \sum_{i=1}^{n} e_{y_i}^T \log[\text{softmax}(f_\theta(x_i))]. \tag{4}$$

In the class imbalanced setting optimizing Eq. 4 is known to produce classifiers that strongly favor common classes and are therefore likely to incorrectly label rare classes during test time. Here we

describe a few methods designed to counteract this phenomenon, which we will use as competitors in our experiments.

## B.2 ERROR RE-WEIGHTING

To counteract the fact that there are fewer terms in the summation in Eq. 4 for rare classes, one may simply weight those terms more greatly. Let $\hat{\mu}_{n,Y}(\cdot) := \mu_n(\mathbb{R}^d \times \cdot)$ be the empirical distribution over the class labels. In error re-weighting, the terms in Eq. 4 are weighted inversely to their class frequency, such that the contribution of a sample in class $c$ to the error decreases with its number of unique examples $n_c$,

$$\mathcal{L}_{\text{re-weighted}}(\mathcal{D}, \theta) := \mathbb{E}_{(X,Y)\sim\mu_n}\left[\frac{\ell(\boldsymbol{e}_Y, \text{softmax}(f_\theta(x_i)))}{\mu_{n,Y}(Y)}\right] \propto -\frac{1}{n}\sum_{i=1}^{n}\frac{a}{n_{y_i}}\boldsymbol{e}_{y_i}^T\log\left[\text{softmax}(f_\theta(x_i))\right],$$
(5)

where $a \in \mathbb{R}$ is a constant to bring the error term back onto the correct scale to avoid vanishing gradient problems or a learning rate $\eta$ that is unusually large, since $n_c \gg 1 \forall c \in \{1, \ldots, C\}$.

## B.3 BATCH-BALANCING

---

**Algorithm 2** Batch-balancing

**Input:** $\mathcal{D}, C, B$                                                           ▷ $C$ is the number of classes, $B$ is the batch size
   $\mathcal{D} = \bigcup_{c=1}^{C}\mathcal{D}_c$                                   ▷ $\mathcal{D}$ is the union of its class partitions
   $\mathcal{D}_c := (x_i^c, y_i^c)_{i=1}^{n_c} \subset \mathcal{X}_c \times \mathcal{Y}_c$   ▷ Each class partition contains $n_c$ ordered pairs
   $[C] = \{1, \ldots, C\}$                                                           ▷ $|[C]| = C$
   **for** $i \in \{1, \ldots, B\}$ **do**
      $c \sim \mathcal{U}([C])$                                             ▷ Sample a class $c$
      $(\tilde{x}_i, \tilde{y}_i) \sim \mathcal{U}(\mathcal{D}_c)$           ▷ Sample an image and label pair from $\mathcal{D}_c$
   **end for**
**Output:** $\tilde{\mathcal{D}} := (\tilde{x}_i, \tilde{y}_i)_{i=1}^{B}$                  ▷ Balanced mini-batch of $B$ image and label pairs

---

In normal batch construction, a sample for a batch is selected uniformly at random from the entire dataset $\mathcal{D}$. Compensating for this by including additional copies of samples from rare classes in the training dataset, or during batch construction, is known as *resampling* or *oversampling* (Chawla et al., 2002; Bellinger et al., 2020). The prototypical version of this selects batch samples by first selecting a class $c \in [C]$, uniformly at random, and then selecting a sample from $\mathcal{D}_c$, uniformly at random. This causes a batch to contain an equal number of samples from each class on average. We term this *batch-balancing*. The works (Ruff et al., 2020; Liznerski et al., 2022) use a slight modification of this where the stochasticity of the labels for each batch is removed so each batch contains an equal number of samples from each class.

The pseudo-code for batch-balancing is described in Alg. 2 in Appx. B.3. Error re-weighting and batch-balancing are not specific to the cross-entropy loss and may be applied to any loss that is the empirical expectation of some loss function. They represent two different ways of remedying class imbalance: one can weigh the rare examples more heavily or one can present the rare examples more often. The following proposition shows that two methods are equivalent in expectation, although we find that batch-balancing always works better in practice (see §5). The sampling distribution for batch-balancing is denoted by $\tilde{\mu}_n$.

**Proposition 1.** *Let $\mathcal{B}$ be a batch selected uniformly at random and $\mathcal{B}'$ be a batch selected using batch-balancing. Then there exists $\lambda > 0$ such that $\lambda\mathbb{E}_{\mathcal{B}}\left[\mathcal{L}_{\text{re-weighted}}(\mathcal{B}, \theta)\right] = \mathbb{E}_{\mathcal{B}'}\left[\mathcal{L}_{\text{vanilla}}(\mathcal{B}', \theta)\right]$ for all $\theta$.*

*Proof.* Let $q = \mathcal{U}([n])$. For the empirical class distribution, $\mu_{n,Y}(Y) = |\{i \mid y_i = y_q\}| / n$. So now we have

$$\mathbb{E}_{(X,Y)\sim\mu_n}\left[\frac{\ell(Y, f_\theta(X))}{\mu_{n,Y}(Y)}\right] = n\mathbb{E}_{q\sim\mathcal{U}([n])}\left[\frac{\ell(y_q, f_\theta(x_q))}{|\{i \mid y_i = y_q\}|}\right]$$

$$= n\sum_{j=1}^{C}\mathbb{E}_{q\sim\mathcal{U}([n])}\left[\frac{\ell(y_q, f_\theta(x_q))}{|\{i \mid y_i = y_q\}|} \mid q = j\right] P_{q\sim\mathcal{U}([n])}(y_q = j)$$

$$= n\sum_{j=1}^{C}\mathbb{E}_{q\sim\mathcal{U}([n])}\left[\frac{\ell(y_q, f_\theta(x_q))}{|\{i \mid y_i = j\}|} \mid q = j\right] \frac{|\{i \mid y_i = j\}|}{n}$$

$$= \sum_{j=1}^{C}\mathbb{E}_{q\sim\mathcal{U}([n])}\left[\ell(y_q, f_\theta(x_q)) \mid y_q = j\right]$$

$$= C\sum_{j=1}^{C}\mathbb{E}_{q\sim\mathcal{U}([n])}\left[\ell(y_q, f_\theta(x_q)) \mid y_q = j\right] C^{-1}. \tag{6}$$

Let $\mathcal{B}$ be the distribution over $[n]$ according to batch-balancing. We know that $\mathcal{U}$ and $\mathcal{B}$ select uniformly, conditioned on class label, so

$$\mathbb{E}_{q\sim\mathcal{U}([n])}\left[\ell(y_q, f_\theta(x_q)) \mid y_q = j\right] = \mathbb{E}_{q\sim\mathcal{B}}\left[\ell(y_q, f_\theta(x_q)) \mid y_q = j\right],$$

and that $\mathcal{B}$ selects class label uniformly

$$P_{q\sim\mathcal{B}}[y_q = j] = C^{-1},$$

for all $j \in [n]$, so Eq. 6 is equal to

$$C\sum_{j=1}^{C}\mathbb{E}_{q\sim\mathcal{U}([n])}\left[\ell(y_q, f_\theta(x_q)) \mid y_q = j\right] C^{-1} = C\sum_{j=1}^{C}\mathbb{E}_{q\sim\mathcal{B}}\left[\ell(y_q, f_\theta(x_q)) \mid y_q = j\right] P_{q\sim\mathcal{B}}[y_q = j]$$

$$= C\mathbb{E}_{q\sim\mathcal{B}}\left[\ell(y_q, f_\theta(x_q))\right]$$

$$= C\mathbb{E}_{(X,Y)\sim\tilde{\mu}_n}\left[\ell(Y, f_\theta(X))\right].$$

$\square$

## C  ANALYZING THE OKO LOSS LANDSCAPE

Here we analyze the optimal network output for the OKO loss and how the loss behaves around such an optimum. Such an optimization landscape analysis is notoriously difficult for non-convex optimization; hence we make some simplifying assumptions. It has been repeatedly observed that well-trained neural networks typically "overfit" the training data. We assume that when a model has memorized its training data, its outputs will be identical for each input corresponding to the same class. In other words, its output on training data only depends on the training label: $f_\theta(x_i) \approx F(y_i) := F_{y_i}$. Here we will consider a matrix $F$ of logit outputs, such that $F_{i,j}$ denotes the logit for class $j$ when the true label is $i$.

One issue with standard cross-entropy is that it strongly encourages the entries of $F$ to diverge: For the cross-entropy risk $\mathcal{R}(F) = \mathbb{E}\left[-e_y^T \log\left(\text{softmax}\left(f(x)\right)\right)\right] = \mathbb{E}\left[-e_y^T \log\left(\text{softmax}\left(F_y\right)\right)\right]$, and for all $F$ and $i, j$, $\frac{\partial \mathcal{R}(F)}{\partial F_{i,i}} > 0$ and for all $i \neq j$, $\frac{\partial \mathcal{R}(F)}{\partial F_{i,j}} < 0$. In other words: standard cross-entropy loss *always* encourages the logits of the true class to move towards $\infty$ and the logits of the wrong class towards $-\infty$. As a result, neural networks tend to be overconfident. In particular, if $f_\theta(x)$ predicts class $\hat{y}$ then $P(\hat{y} \mid x) < [\text{softmax}(f_\theta(x))]_{\hat{y}}$.

A natural way to counteract this issue is via weight decay. Indeed, weight decay has been shown to improve calibration. However, this improvement comes at a cost of generalization, and modern networks therefore typically utilize little to no weight decay (Guo et al., 2017). Thus, there is a desire to find ways to calibrate networks without using weight decay. *Label smoothing* is one potential solution to this since it encourages $\exp F_y$ to be proportional to $e_y(1 - \alpha) + \alpha/C$ (Carratino et al., 2022). We include label smoothing as a competitor for our method in the experimental section.

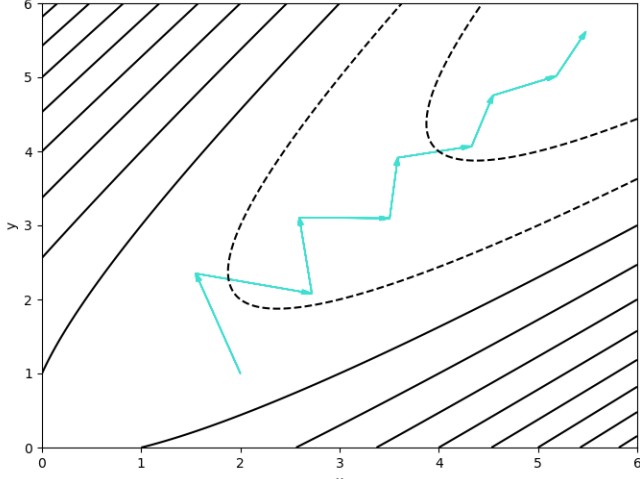

Figure 6: This figure shows the contour lines and gradient descent path of $f(x, y) = -x - y + (y - x)^2$. Note that for fixed $x$ or $y$, $f$ is convex in the other variable. While gradient descent does indeed diverge to $(x, y) \to (\infty, \infty)$ the properties of $f$ slow its divergence.

Before deriving the risk functions for the hard and soft OKO losses, we must introduce further notation. Let $\mathbb{Y}_i = \left\{ S \mid S \in 2^{[C]}, |S| = k, i \notin S \right\}$. $\mathbb{Y}_i$ represents a potential set of odd labels when $y' = i$. For OKO we have

$$\mathcal{R}_{\text{soft}}(F) = -\sum_{i=1}^{C} \sum_{Y \in \mathbb{Y}_i} \sum_{j=1}^{k+2} \log \left[ \text{softmax} \left( \sum_{\ell=1}^{k+2} F_{Y'_\ell} \right) \right]_{Y'_j} \tag{7}$$

$$\mathcal{R}_{\text{hard}}(F) = -\sum_{i=1}^{C} \sum_{Y \in \mathbb{Y}_i} \log \left[ \text{softmax} \left( \sum_{\ell=1}^{k+2} F_{Y'_\ell} \right) \right]_i. \tag{8}$$

See Appx. C for a derivation of these. The following proposition demonstrates that the OKO risks naturally encourage the risk not to overfit by constraining when each index of $F$ is viewed individually.

**Proposition 2.** *For a fixed $i, j$, both $\mathcal{R}_{\text{soft}}(F)$ and $\mathcal{R}_{\text{hard}}(F)$ are convex and each admits a unique global minimum with respect to $F_{i,j}$.*

This acts as an implicit form of smoothing and helps to keep the logits from diverging during training. This is because the gradient descent path for functions like those described in Proposition 2 tend to meander rather than directly diverge. A toy example of this phenomenon can be found in Figure 6. Unlike label smoothing (Müller et al., 2019; Carratino et al., 2022), however, $F$ is not encouraged to converge to a unique minimum. Optimizing the (hard) OKO risk still allows $F$ to diverge if that's advantageous — as we demonstrate in the following proposition.

**Proposition 3.** *There exist an initial value, $F^{(0)}$, such that the sequence of points given by minimizing $\mathcal{R}_{\text{hard}}$ with fixed step size gradient descent, $F^{(0)}, F^{(1)}, F^{(2)}, \ldots$, yields, $\lim_{a \to \infty} F_{i,i}^{(a)} \to \infty$ and $\lim_{a \to \infty} F_{i,j}^{(a)} \to -\infty$, for all $i$ and $j \neq i$.*

Hence, the OKO risk strikes a balance between the excessive overconfidence caused by standard cross-entropy and the inflexible calibration of fixed minima in label smoothing (Carratino et al., 2022).

Here we derive the expressions in Eq. 7 and Eq. 8 in the main text. We have that

$$
\mathbb{E}_{\mathcal{S} \sim \mathscr{A}}\left[\ell_{\text{oko}}^{\text{soft}}\left(\mathcal{S}_y, f_\theta\left(\mathcal{S}_x\right)\right)\right] = \sum_{i=1}^{k} P\left(y' = i\right) \mathbb{E}_{\mathcal{S} \sim \mathscr{A}}\left[\ell_{\text{oko}}^{\text{soft}}\left(\mathcal{S}_y, f_\theta\left(\mathcal{S}_x\right)\right) \mid y' = i\right]
$$

$$
= \sum_{i=1}^{k} P\left(y' = i\right) \sum_{Y \in \mathbb{Y}_i}\left(P\left(\left\{y'_3, .., y'_{k+2}\right\} = Y \mid y' = i\right)\right. \tag{9}
$$

$$
\left. \cdots \times \mathbb{E}_{\mathcal{S} \sim \mathscr{A}}\left[\ell_{\text{oko}}^{\text{soft}}\left(\mathcal{S}_y, f_\theta\left(\mathcal{S}_x\right)\right) \mid y' = i, \left\{y'_3, .., y'_{k+2}\right\} = Y\right]\right), \tag{10}
$$

noting that the parenthesis after the second summation in Eq. 9 extends to the end of Eq. 10. Since $y'$ is chosen uniformly at random $P\left(y' = i\right)$ are equal for all $i$ and similarly $\left(y'_3, \ldots, y'_{k+2}\right)$ are chosen uniformly at random given $y'$ and $|\mathbb{Y}_i|$ are all equal so $P\left(\left\{y'_3, \ldots, y'_{k+2}\right\} = Y \mid y' = i\right)$ are equal for all $Y$ and $i$, thus we can ignore these terms when optimizing over $F$. Letting $Y' = (i, i, Y_1, \ldots, Y_k)$ in the summation we have, that $\mathbb{E}_{\mathcal{S} \sim \mathscr{A}}\left[\ell_{\text{oko}}^{\text{soft}}\left(\mathcal{S}_y, f_\theta\left(\mathcal{S}_x\right)\right)\right]$ is proportional to,

$$
\sum_{i=1}^{k} \sum_{Y \in \mathbb{Y}_i} \mathbb{E}_{\mathcal{S} \sim \mathscr{A}}\left[\ell_{\text{oko}}^{\text{soft}}\left(\mathcal{S}_y, f_\theta\left(\mathcal{S}_x\right)\right) \mid y' = i, \left\{y'_3, \ldots, y'_{k+2}\right\} = Y\right]
$$

$$
= \sum_{i=1}^{k} \sum_{Y \in \mathbb{Y}_i} \mathbb{E}_{\mathcal{S} \sim \mathscr{A}}\left[\left((k+2)^{-1} \sum_{i=1}^{k+2} \boldsymbol{e}_{y'_i}\right)^T \log\left[\text{softmax}\left(f_\theta\left(\mathcal{S}_x\right)\right)\right] \mid y' = i, \left\{y'_3, \ldots, y'_{k+2}\right\} = Y\right]
$$

$$
\propto \sum_{i=1}^{k} \sum_{Y \in \mathbb{Y}_i} \mathbb{E}_{\mathcal{S} \sim \mathscr{A}}\left[\left(\sum_{i=1}^{k+2} \boldsymbol{e}_{Y'_i}\right)^T \log\left[\text{softmax}\left(\sum_{i=1}^{k+2} F_{Y'_i}\right)\right]\right]
$$

$$
= \sum_{i=1}^{k} \sum_{Y \in \mathbb{Y}_i}\left(\sum_{i=1}^{k+2} \boldsymbol{e}_{Y'_i}\right)^T \log\left[\text{softmax}\left(\sum_{i=1}^{k+2} F_{Y'_i}\right)\right]
$$

$$
= \sum_{i=1}^{k} \sum_{Y \in \mathbb{Y}_i} \sum_{j=1}^{k+2} \log\left[\text{softmax}\left(\sum_{\ell=1}^{k+2} F_{Y'_\ell}\right)\right]_{Y'_j}. \tag{11}
$$

The derivation of the hard risk is similar, however the third summation in Eq. 11 only contains the $i$ term, for the single hard label.

## C.1 PROOFS

Before proving Proposition 2 we will first introduce the following support lemma, which will be proven later.

**Lemma 3.** *Let, $N, N' \in \mathbb{N}$ be positive. Let $q_i, q'_i \in \mathbb{R}^C$, with $q_{i,1} = a_i x + b_i$, $q'_{i,1} = a'_i x + b'_i$ (the remaining entries of $q_i$ and $q'_i$ are fixed and do not depend on $x$) with $a_i > 0$ and $a'_i > 0$, and $n_i$ be a sequence of $N'$ elements in $[C] \setminus \{1\}$. Then*

$$
f(x) = \underbrace{-\sum_{i=1}^{N} \log\left(\text{softmax}\left(q_i\right)\right)_1}_{\mathscr{L}} \underbrace{-\sum_{i=1}^{N'} \log\left(\text{softmax}\left(q'_i\right)\right)_{n_i}}_{\mathscr{R}} \tag{12}
$$

*is strictly convex and admits a unique minimizer.*

*Proof of Proposition 2.* This proof will be proven using surrogate indices $i', j'$ in place of $i, j$ in the proposition for $F_{i,j}$; it will be useful to be able to use $i$ and $j$ to refer to indices in other expressions.

Observe that simply relabeling the network outputs does not affect the risk so, for a permutation $\sigma$ and $G$ defined by $G_{i',j'} = F_{\sigma(i'),\sigma(j')}$ we have that $\mathcal{R}(F) = \mathcal{R}(G)$. Because of this we will simply let $j' = 1$ for concreteness. We will begin with the case where $i' = j'$.

**Case $i' = j' = 1$:** Note that each summand in Eq. 7 and Eq. 8 is either a constant with respect to $F_{1,1}$ or has the form of the left hand sum, $\mathscr{L}$, or right hand sum, $\mathscr{R}$, of Eq. 12, substituting in $x \leftarrow F_{1,1}$ in the statement of Lemma 3. To finish the $i' = j'$ case we will show that both Eq. 7 and Eq. 8 has a summand of the form in $\mathscr{L}$ and one summand of the form in $\mathscr{R}$:

- For $\mathscr{L}$: Consider $i = 1$ and an arbitrary $Y \in \mathbb{Y}_1$ for Eq. 8; for Eq. 7 we use the same values with $j = 1$ since $Y'_1 = 1$.

- For $\mathscr{R}$: In Eq. 7 we can let $i = 1$, $Y \in \mathbb{Y}_1$ be arbitrary, and $j = 3$ since $Y'_3 \neq 1 = j'$ in that case. For Eq. 8 we need only consider $i = 2$ and some $Y = \mathbb{Y}_2$ that contains an entry with 1.

From Lemma 3 it follows that both $\mathcal{R}_{\text{soft}}(F)$ and $\mathcal{R}_{\text{hard}}(F)$ are strictly convex and contain a unique minimum when optimizing over $F_{i',i'}$.

**Case $i' \neq j' = 1$:** This case proceeds in a similar fashion to the last.

- For $\mathscr{L}$: In Eq. 8 we have $i = 1$ and some $Y \in \mathbb{Y}_1$, so that $Y'_3 = i'$; for Eq. 7 we add $j = 1$ so $Y'_1 = 1$.

- For $\mathscr{R}$: In Eq. 8 we have $i = 2$, $Y \in \mathbb{Y}_2$ such that $i'$ is in $Y$. We can use the same values for $i$ and $Y$ for Eq. 7, with $j = 1$ since $Y_1 = 2$.

$\square$

*Proof of Lemma 3.* To show strict convexity, we will show that $\frac{d^2 f}{dx^2}$ is strictly positive. First, we have that

$$
\frac{df}{dx} = -\sum_{i=1}^{N} a_i \left(1 - \text{softmax}\,(q_i)_1\right) - \sum_{i=1}^{N'} a'_i \left(-\text{softmax}\,(q'_i)_1\right)
$$

$$
= \sum_{i=1}^{N} a_i \left(\text{softmax}\,(q_i)_1 - 1\right) + \sum_{i=1}^{N'} a'_i \left(\text{softmax}\,(q'_i)_1\right) \tag{13}
$$

and thus

$$
\frac{d^2 f}{dx^2} = \sum_{i=1}^{N} a_i^2 \left(1 - \text{softmax}\,(q_i)_1\right) \text{softmax}\,(q_i)_1 + \sum_{i=1}^{N'} a_i'^2 \left(\text{softmax}\,(q'_i)_1\right) \left(1 - \text{softmax}\,(q'_i)\right)
$$

which is clearly positive for all $x$. To demonstrate the existence of a minimizer we will show that $\frac{df}{dx}$ attains both positive and negative values as a function of $x$ and, by the intermediate value theorem, $\frac{df}{dx}$ must equal 0 somewhere. To see this observe that

$$
\lim_{x \to \infty} \sum_{i=1}^{N} a_i \left(\text{softmax}\,(q_i)_1 - 1\right) + \sum_{i=1}^{N'} a'_i \left(\text{softmax}\,(q'_i)_1\right) = \sum_{i=1}^{N'} a'_i > 0
$$

$$
\lim_{x \to -\infty} \sum_{i=1}^{N} a_i \left(\text{softmax}\,(q_i)_1 - 1\right) + \sum_{i=1}^{N'} a'_i \left(\text{softmax}\,(q'_i)_1\right) = \sum_{i=1}^{N} -a_i < 0,
$$

which completes the proof. $\square$

*Proof of Proposition 3.* Let $\mathcal{F} \subset \mathbb{R}^{C \times C}$ be the set of matrices $F$ where $F_{i,i} = F_{j,j}$ for all $i, j$, and $F_{i,j} = F_{i',j'}$ for all $i \neq j$, $i' \neq j'$. Let $F(a, b) \in \mathcal{F}$ be the matrix which contains $a$ in the diagonal entries and $b$ in all other entries. The chain rule tells us that

$$
\frac{\partial}{\partial a} \mathcal{R}_{\text{hard}}\left(F(a, b)\right) = \sum_{i=1}^{C} \frac{\partial}{\partial F_{i,i}} \mathcal{R}_{\text{hard}}\left(F\right) \bigg|_{F = F(a,b)}, \tag{14}
$$

the sum of all the partial derivatives along the diagonal, and

$$\frac{\partial}{\partial b}\mathcal{R}_{\text{hard}}\left(F(a,b)\right) = \sum_{i \neq j} \frac{\partial}{\partial F_{i,j}}\mathcal{R}_{\text{hard}}\left(F\right)\bigg|_{F=F(a,b)}, \tag{15}$$

the sum of all partial derivatives for the entries off the diagonal. Due to the invariance with respect to labeling, as in the proof of Proposition 2, for any $F \in \mathcal{F}$ it follows that $\frac{\partial}{\partial F_{i,i}}\mathcal{R}_{\text{hard}}(F) = \frac{\partial}{\partial F_{i',i'}}\mathcal{R}_{\text{hard}}(F)$ for all $i$ and $i'$, and $\frac{\partial}{\partial F_{i,j}}\mathcal{R}_{\text{hard}}(F) = \frac{\partial}{\partial F_{i',j'}}\mathcal{R}_{\text{hard}}(F)$ for all $i \neq j$, $i' \neq j'$. Because of this $\nabla\mathcal{R}_{\text{hard}}\left(F(a,b)\right)$ will always lie in $\mathcal{F}$ and thus a path following gradient descent starting in $\mathcal{F}$ will always remain in $\mathcal{F}$.

Considering Eq. 14 and Eq. 15, if we show that $-\frac{\partial}{\partial a}\mathcal{R}_{\text{hard}}\left(F(a,b)\right) > 0$ and $-\frac{\partial}{\partial b}\mathcal{R}_{\text{hard}}\left(F(a,b)\right) < 0$, for any $a,b$, it would follow that, for $F \in \mathcal{F}$, $-\nabla\mathcal{R}_{\text{hard}}\left(F\right) = F(a',b')$ for some $a' > 0$ and $b' < 0$. This would imply that gradient descent starting from an element of $\mathcal{F}$ will diverge to $F\left(\infty, -\infty\right)$, which would complete the proof. We will now proceed proving $-\frac{\partial}{\partial a}\mathcal{R}_{\text{hard}}\left(F(a,b)\right) > 0$ and $-\frac{\partial}{\partial b}\mathcal{R}_{\text{hard}}\left(F(a,b)\right) < 0$.

The risk expression applied to $F(a,b)$ is equal to

$$\mathcal{R}_{\text{hard}}(F(a,b)) = -\sum_{i=1}^{k}\sum_{Y \in \mathbb{Y}_i} \log\left[\text{softmax}\left(\sum_{\ell=1}^{k+2} F(a,b)_{Y'_\ell}\right)\right]_i. \tag{16}$$

For concreteness we will consider the summand with $i = 1$ and $Y = [2, \dots, k+1]$ fixed, which implies $Y' = [1, 1, 2, \dots, k+1]$ is also fixed. In this case we have that

$$\sum_{\ell=1}^{k+2} F(a,b)_{Y'_\ell} = \begin{bmatrix} 2a + kb \\ a + (k+1)b \\ \vdots \\ a + (k+1)b \\ (k+2)b \\ \vdots \\ (k+2)b \end{bmatrix},$$

with $k$ entries containing $a + (k+1)b$ and $C - (k+1)$ entries containing $(k+2)b$ (note that $C - (k+1)$ is nonnegative). Continuing with fixed $i$ and $Y'$ have that

$$\log\left[\text{softmax}\left(\sum_{\ell=1}^{k+2} F(a,b)_{Y'_\ell}\right)\right]_i$$
$$= \log\left(\frac{\exp\left(2a + bk\right)}{\exp\left(2a + bk\right) + k\exp\left(a + b(k+1)\right) + (C - k - 1)\exp\left((k+2)b\right)}\right). \tag{17}$$

We will define $R(a,b)$ to be equal to Eq. 17 Note that every summand in Eq. 16 is equal to Eq. 17. Because of this we need only show that $\frac{\partial}{\partial a}R(a,b) > 0$ and $\frac{\partial}{\partial b}R(a,b) < 0$ to finish the proof.

Differentiating with respect to $a$ gives us

$$\frac{\partial}{\partial a}R(a,b) = 2 - \frac{2\exp\left(2a + bk\right) + k\exp\left(a + b(k+1)\right)}{\exp\left(2a + bk\right) + k\exp\left(a + b(k+1)\right) + (C - k - 1)\exp\left((k+2)b\right)}.$$

Letting

$$Q(a,b) := \exp\left(2a + bk\right) + k\exp\left(a + b(k+1)\right) + (C - k - 1)\exp\left((k+2)b\right)$$

it follows that

$$\frac{\partial}{\partial a}R(a,b) = 2 - \frac{\exp\left(2a + bk\right) + k\exp\left(a + b(k+1)\right)}{Q(a,b)} - \frac{\exp\left(2a + bk\right)}{Q(a,b)}.$$

We have that

$$\frac{\exp\left(2a + bk\right) + k\exp\left(a + b(k+1)\right)}{Q(a,b)} \leq 1 \text{ and } \frac{\exp\left(2a + bk\right)}{Q(a,b)} < 1$$

so $\frac{\partial}{\partial a} R(a, b) > 0$.

Differentiating with respect to $b$ we get

$$\frac{\partial}{\partial b} R(a, b)$$
$$= k - \frac{k \exp\left(2a + bk\right) + k(k+1) \exp\left(a + b(k+1)\right) + (k+2)(C - k - 1) \exp\left((k+2)b\right)}{\exp\left(2a + bk\right) + k \exp\left(a + b(k+1)\right) + (C - k - 1) \exp\left((k+2)b\right)}.$$

Observe that

$$k < \frac{k \exp\left(2a + bk\right) + k(k+1) \exp\left(a + b(k+1)\right) + (k+2)(C - k - 1) \exp\left((k+2)b\right)}{\exp\left(2a + bk\right) + k \exp\left(a + b(k+1)\right) + (C - k - 1) \exp\left((k+2)b\right)}$$

so $\frac{\partial}{\partial b} R(a, b) < 0$, which completes the proof. $\qquad\square$

## D  PROOF OF THEOREM 1

To prove this let $\mathbb{F}_0 \subset \mathbb{F}$ be such that $f \in \mathbb{F}_0$ satisfies $f(i)_1 = 0$ for all $i$. We have the following lemma showing that $\mathbb{F}_0$ can be used in place of $\mathbb{F}$ in our proof.

**Lemma 4.** *Let $v_1, \ldots, v_M$ be in $\mathbb{R}^d$. Then there exists $v'_1, \ldots, v'_M$ with $v'_{i,1} = 0$ for all $i$, such that, for all $w_1, \ldots, w_M$ in $\mathbb{R}$, the following holds*

$$\mathrm{softmax}\left(\sum_{i=1}^{M} w_i v_i\right) = \mathrm{softmax}\left(\sum_{i=1}^{M} w_i v'_i\right).$$

This essentially allows us avoid the complications arising from that fact that there exists an uncountable infinitude of distinct vectors, in particular $f(a)$ with various $a \in \mathbb{R}$ and $f(a)_1 = a$, such that $\mathrm{softmax}(f(a)) = \mathrm{softmax}(f(a'))$ when $a \neq a'$, by simply selecting the unique representative with $f_1 = 0$.

*Proof of Lemma 4.* First we will show for all $x \in \mathbb{R}^d$ and $c \in \mathbb{R}$ that $\mathrm{softmax}(x) = \mathrm{softmax}\left(x + c\vec{\mathbb{1}}\right)$, where $\vec{\mathbb{1}} \in \mathbb{R}^d$ is the ones vector. Let $j \in [d]$ be an arbitrary index. We will show that $\mathrm{softmax}\left(x + c\vec{\mathbb{1}}\right)_j = \mathrm{softmax}(x)_j$. Observe that

$$\mathrm{softmax}\left(x + c\vec{\mathbb{1}}\right)_j = \frac{\exp\left(x_j + c\right)}{\sum_{k=1}^{d} \exp\left(x_k + c\right)}$$
$$= \frac{\exp\left(x_j\right) \exp\left(c\right)}{\sum_{k=1}^{d} \exp\left(x_k\right) \exp\left(c\right)}$$
$$= \frac{\exp\left(c\right) \exp\left(x_j\right)}{\exp\left(c\right) \sum_{k=1}^{d} \exp\left(x_k\right)}$$
$$= \frac{\exp\left(x_j\right)}{\sum_{k=1}^{d} \exp\left(x_k\right)}$$
$$= \mathrm{softmax}(x)_j.$$

Because the above equality is true for all indices $j$, it holds for the whole vector, so $\mathrm{softmax}(x) = \mathrm{softmax}\left(x + c\vec{\mathbb{1}}\right)$.

Let $v_i' = v_i - v_{i,1}\vec{\mathbb{1}}$ for all $i$. Note that $v_{i,1}' = 0$ for all $i$. Now we have that

$$\text{softmax}\left(\sum_{i=1}^{M} w_i v_i'\right) = \text{softmax}\left(\sum_{i=1}^{M} w_i \left(v_i - v_{i,1}\vec{\mathbb{1}}\right)\right)$$

$$= \text{softmax}\left(\sum_{i=1}^{M} w_i v_i - w_i v_{i,1}\vec{\mathbb{1}}\right)$$

$$= \text{softmax}\left(\left(\sum_{i=1}^{M} w_i v_i\right) + \left(-\sum_{i=1}^{M} w_i v_{i,1}\vec{\mathbb{1}}\right)\right)$$

$$= \text{softmax}\left(\sum_{i=1}^{M} w_i v_i\right),$$

where the last line follows from the equality shown at the beginning of this proof. $\square$

Using this we can prove the main theorem.

*Proof of Theorem 1.* From Lemma 4 the theorem statement is true iff it holds for minimizers in $\mathbb{F}_0$, so we will prove the theorem for $f$ where $f(\cdot)_1 = 0$.

Let $Q_\epsilon(a_0, a_1, a_2) = Q_\epsilon(a) \triangleq \mathbb{E}_{\mathcal{S} \sim \mathscr{A}} \left[\ell_{\text{oko}}^{\text{hard}}\left(\mathcal{S}_y, f\left(\mathcal{S}_x\right)\right)\right]$, with $f$ satisfying $f(0) = [0, a_0]$, $f(1) = [0, a_1]$, and $f(2) = [0, a_2]$. For the remainder for this proof we will denote the entries of vectors like $a$ beginning with 0, as above, rather than 1.

To begin note that

$$Q_\epsilon(a) = \sum_{i=1}^{16} -p_{\epsilon,i} \log\left(V_i(a)\right),$$

where the $V_i$ are defined in the following table (Table 4).

| # | $y_1$ | $y_2$ | $y_3$ | $x_1$ | $x_2$ | $x_3$ | $P_{\mathscr{A}_\epsilon}\left(x_1, x_2, x_3, y_1, y_2, y_3\right) = p_{\epsilon,\#}$ | $V_\#(a)$ |
|---|---|---|---|---|---|---|---|---|
| 1 | 1 | 1 | 2 | 0 | 0 | 0 | $(1-\epsilon)^3/2$ | $\frac{\exp(0)}{\exp(0)+\exp(3a_0)}$ |
| 2 | 2 | 2 | 1 | 0 | 0 | 0 | $(1-\epsilon)^3/2$ | $\frac{\exp(3a_0)}{\exp(0)+\exp(3a_0)}$ |
| 3 | 1 | 1 | 2 | 0 | 0 | 2 | $\epsilon(1-\epsilon)^2/2$ | $\frac{\exp(0)}{\exp(0)+\exp(2a_0+a_2)}$ |
| 4 | 1 | 1 | 2 | 1 | 0 | 0 | $\epsilon(1-\epsilon)^2/2$ | $\frac{\exp(0)}{\exp(0)+\exp(2a_0+a_1)}$ |
| 5 | 1 | 1 | 2 | 0 | 1 | 0 | $\epsilon(1-\epsilon)^2/2$ | $\frac{\exp(0)}{\exp(0)+\exp(2a_0+a_1)}$ |
| 6 | 2 | 2 | 1 | 0 | 0 | 1 | $\epsilon(1-\epsilon)^2/2$ | $\frac{\exp(2a_0+a_1)}{\exp(0)+\exp(2a_0+a_1)}$ |
| 7 | 2 | 2 | 1 | 2 | 0 | 0 | $\epsilon(1-\epsilon)^2/2$ | $\frac{\exp(2a_0+a_2)}{\exp(0)+\exp(2a_0+a_2)}$ |
| 8 | 2 | 2 | 1 | 0 | 2 | 0 | $\epsilon(1-\epsilon)^2/2$ | $\frac{\exp(2a_0+a_2)}{\exp(0)+\exp(2a_0+a_2)}$ |
| 9 | 1 | 1 | 2 | 1 | 1 | 0 | $\epsilon^2(1-\epsilon)/2$ | $\frac{\exp(0)}{\exp(0)+\exp(a_0+2a_1)}$ |
| 10 | 1 | 1 | 2 | 1 | 0 | 2 | $\epsilon^2(1-\epsilon)/2$ | $\frac{\exp(0)}{\exp(0)+\exp(a_0+a_1+a_2)}$ |
| 11 | 1 | 1 | 2 | 0 | 1 | 2 | $\epsilon^2(1-\epsilon)/2$ | $\frac{\exp(0)}{\exp(0)+\exp(a_0+a_1+a_2)}$ |
| 12 | 2 | 2 | 1 | 2 | 2 | 0 | $\epsilon^2(1-\epsilon)/2$ | $\frac{\exp(a_0+2a_2)}{\exp(0)+\exp(a_0+2a_2)}$ |
| 13 | 2 | 2 | 1 | 2 | 0 | 1 | $\epsilon^2(1-\epsilon)/2$ | $\frac{\exp(a_0+a_1+a_2)}{\exp(0)+\exp(a_0+a_1+a_2)}$ |
| 14 | 2 | 2 | 1 | 2 | 1 | 0 | $\epsilon^2(1-\epsilon)/2$ | $\frac{\exp(a_0+a_1+a_2)}{\exp(0)+\exp(a_0+a_1+a_2)}$ |
| 15 | 1 | 1 | 2 | 1 | 1 | 2 | $\epsilon^3/2$ | $\frac{\exp(0)}{\exp(0)+\exp(2a_1+a_2)}$ |
| 16 | 2 | 2 | 1 | 2 | 2 | 1 | $\epsilon^3/2$ | $\frac{\exp(a_1+2a_2)}{\exp(0)+\exp(a_1+2a_2)}$ |

Table 4: Terms in expectation

From this it is clear that $Q_\epsilon$ is continuous for all $\epsilon$.

We will now show that for all $\epsilon \in (0, 1)$ there exists a minimizer $a_\epsilon \in \arg\min_a Q_\varepsilon(a)$ by contradiction. For sake of contradiction suppose there exists $\epsilon'$ for which no such minimizer exists. Because $Q_{\epsilon'}$ is bounded from below we can can find a sequence $(a_i)_{i=1}^\infty$ such that $Q_{\epsilon'}(a_i) \to \inf_a Q_{\epsilon'}(a)$. Because $\sum_{i=1}^2 -p_{\epsilon',i} \log(V_i(a)) \to \infty$ as $|a_0| \to \infty$ (here $a_0$ is the first entry of $a$, rather than one vector from the sequence $a_i$, incurring a slight abuse of notation) it follows that the sequence $(a_{i,0})_{i=1}^\infty$ must be bounded and thus contains a convergent subsequence. We will now assume that $(a_i)_{i=1}^\infty$ is such a sequence. Note that $\sum_{i=3}^8 -p_{\epsilon',i} \log(V_i(a_j))$ would diverge were $a_{j,1}$ or $a_{j,2}$ to diverge (because $a_{j,0}$ remains bounded) and, because every term $-p_{\epsilon,i} \log(V_i(a)) > 0$, $(a_i)_{i=1}^\infty$ must remain bounded otherwise $Q_\epsilon(a_j)$ would diverge. Thus $a_i$ has a convergent subsequence and its limit must be a minimizer by continuity, a contradiction. Therefore minimizers $a_\epsilon$ exist.

Let $a^\star = [0, -\log(2), \log(2)]$ and define

$$Q_1 : a \mapsto \sum_{i=1}^2 -p_{\epsilon,i} \log(V_i(a))$$

$$Q_2 : a \mapsto \sum_{i=3}^8 -p_{\epsilon,i} \log(V_i(a))$$

$$Q_3 : a \mapsto \sum_{i=9}^{16} -p_{\epsilon,i} \log(V_i(a)),$$

with the $\epsilon$ subscript for $Q$ left implicit to simplify notation. Observe that $a_0^\star$ minimizes $Q_1$, which holds iff $a_0^\star = 0$, and that $a^\star$ minimizes $Q_2$ with $a_0^\star$ fixed to 0. Showing that these are minima follows from simple calculus and algebra. The functions $Q_1$ and $Q_2$ can be shown to be strictly convex by showing that the second derivative/Hessian is positive/positive-definite. The minima can be found by simply taking the derivative/gradient of these terms and setting them equal to zero. All instances in this proof where we assert strict convexity and find minima follow from this kind of argument.

We will now show that $a_{\epsilon,0} \to 0$ by contradiction. All limits are with respect to $\epsilon \to 0$. For the sake of contradiction assume there is a decreasing subsequence $(\epsilon_i)_{i=1}^\infty$ which converges to 0 and $\delta > 0$ such that $|a_{\epsilon_i,0} - a_0^\star| \geq \delta$. Since $a \mapsto \sum_{i=1}^2 -\log(V_i(a))$ is strictly convex in its first index and is minimized when $a_0 = 0$, there exists $\Delta_1 > 0$ such that $\sum_{i=1}^2 -\log(V_i(a_{\epsilon_j})) - -\log(V_i(a^\star)) \geq \Delta_1$, for all $j$ (we've left in the double negative to emphasize that we are subtracting the convex term). Now we have that

$$Q_{\epsilon_i}(a_{\epsilon_i}) - Q_{\epsilon_i}(a^\star) \geq \frac{(1-\epsilon_i)^3}{2} \Delta_1 - \sum_{j=2}^3 Q_j(a^\star)$$

$$= \frac{(1-\epsilon_i)^3}{2} \Delta_1 - O(\epsilon_i)$$

and thus, for sufficiently large $i$, $Q_{\epsilon_i}(a_{\epsilon_i}) > Q_{\epsilon_i}(a^\star)$, a contradiction. We have now shown that $a_{\epsilon,0} \to 0$.

We will now show that $a_{\epsilon,1}$ and $a_{\epsilon,2}$ must be bounded, again via contradiction. As before we can consider the case where $\epsilon_i$ converges to zero with $a_{\epsilon_i,1}$ diverging (the proof is virtually identical for $a_{\epsilon_i,2}$). Note that a minimizer of $Q_\epsilon$ is also a minimizer of $\epsilon^{-1} Q_\epsilon$. Now we have

$$\epsilon_i^{-1} Q_{\epsilon_i}(a_{\epsilon_i}) - \epsilon_i^{-1} Q_{\epsilon_i}(a^\star)$$

$$= \underbrace{\epsilon_i^{-1} Q_1(a_{\epsilon_i}) - \epsilon_i^{-1} Q_1(a^\star)}_{\geq 0} + \epsilon_i^{-1} Q_2(a_{\epsilon_i}) - \epsilon_i^{-1} Q_2(a^\star) + \underbrace{\epsilon_i^{-1} Q_3(a_{\epsilon_i})}_{\geq 0} - \underbrace{\epsilon_i^{-1} Q_3(a^\star)}_{O(\epsilon_i)}$$

$$\geq \epsilon_i^{-1} Q_2(a_{\epsilon_i}) - \epsilon_i^{-1} Q_2(a^\star) + O(\epsilon_i). \tag{18}$$

Observe that if $a_{\epsilon_i,1}$ diverges with $a_{\epsilon_i,0} \to 0$ then the first term in the last line goes to $\infty$ (see terms 4 and 6 in Table 4) which would contradict the optimality of $a_{\epsilon_i}$ due to Eq. 18. Thus we have that $a_\epsilon$ must remain bounded as $\epsilon \to 0$.

Since $a_\epsilon$ is bounded, there exists yet another sequence $\epsilon_i$ such that $a_{\epsilon_i}$ converges. We will call this limit $a'$ noting that $a'_0 = 0$. Again we have

$$
\begin{aligned}
&\epsilon_i^{-1} Q_{\epsilon_i}(a_{\epsilon_i}) - \epsilon_i^{-1} Q_{\epsilon_i}(a^\star) \\
&= \underbrace{\epsilon_i^{-1} Q_1(a_{\epsilon_i}) - \epsilon_i^{-1} Q_1(a^\star)}_{\geq 0} + \epsilon_i^{-1} Q_2(a_{\epsilon_i}) - \epsilon_i^{-1} Q_2(a^\star) + \underbrace{Q_3(a_{\epsilon_i})}_{\geq 0} - \underbrace{\epsilon_i^{-1} Q_3(a^\star)}_{O(\epsilon_i)} \\
&\geq \epsilon_i^{-1} Q_2(a_{\epsilon_i}) - \epsilon_i^{-1} Q_2(a^\star) - \underbrace{\epsilon_i^{-1} Q_3(a^\star)}_{O(\epsilon_i)} \\
&= \frac{(1-\epsilon_i)^2}{2} \left( \sum_{j=3}^{8} -\log\left(V_j\left(a_{\epsilon_i}\right)\right)) - -\log\left(V_j\left(a^\star\right)\right) \right) - O(\epsilon_i).
\end{aligned}
\tag{19}
$$

Note that the interior summation of the last line converges to $\sum_{i=3}^{8} -\log\left(V_i\left(a'\right)\right) - -\log\left(V_i\left(a^\star\right)\right)$ which must equal zero since $\sum_{i=3}^{8} -\log\left(V_i\left(\cdot\right)\right)$ is continuous and otherwise the optimality of $a_{\epsilon_i}$ would be violated in Eq. 19 for sufficiently large $i$. Finally note that $\sum_{i=3}^{8} -\log\left(V_i\left(\cdot\right)\right)$ is strictly convex and thus $a' = a^\star$. Since $a_\epsilon$ is bounded for small $\epsilon$ and because every convergent subsequence $\epsilon_i \to 0$ causes $a_{\epsilon_i} \to a^\star$, it follows that $a_\epsilon \to a^\star$. $\qquad\square$

## E  CALIBRATION

Here, we provide more intuition about our new entropy-based measure of datapoint calibration. In a sense, our measure is a normalized scoring rule that provides localized probabilistic insight.

### E.1  BACKGROUND

Rare classes make it difficult for calibration in practice, that is the predicted probabilities match the true probabilities of an event occurring, essential for making reliable decisions based on the model's predictions. Specifically, if $f : \mathcal{X} \to [0, 1]$ is a probabilistic model for binary labels, we want $\mathbb{E}[y|f(x) = v] = v$. For multi-class labels, this generalizes for each class such that for predicted probability vector $p \in [0, 1]^C$, $\mathbb{E}[e_y|f(x) = p] = p$.

There are also various ways to measure calibration error. The most common is the expected calibration error; however, arbitrarily small perturbations to the predictor $f$ can cause large fluctuations in $ECE(f)$ (Kakade & Foster, 2004; Foster & Hart, 2018). As a simple example, consider the uniform distribution over a two-point space $\mathcal{X} = \{x_1, x_2\}$, where the labels are $y_1 = 0, y_2 = 1$ respectively. The predictor f which simply predicts 1/2 is perfectly calibrated, so $ECE(f) = 0$. However, a more accurate estimator $f(x_1) = 1/2 - \epsilon$ and $f(x_2) = 1/2 + \epsilon$ for any small $\epsilon > 0$ suffers calibration error $ECE(f) = 1/2 - \epsilon$. Note that this discontinuity also presents a barrier to popular heuristics for estimating the ECE. However, it has been show that ECE and most binning based variants give an upper bound on the true distance to calibration (Błasiok et al., 2022). It has been shown that sample-efficient continuous, complete and sound calibration measures do exist and they all generally measure a distance to the most calibrated predictor, and many of them are similar to binned ECE. Therefore, we utilize binned ECE and brier score as two notions of calibration.

In the multiclass setting, while there is not a consensus on how to measure calibration error, we use the most common way to extend calibration metrics, which is widely implemented via the one-vs-all method (Zadrozny & Elkan, 2002). Recently, there were other multi-class calibration measures proposed, such as class-wise calibration (Kull et al., 2019) and decision calibration (Zhao et al., 2021).

**Scoring Rules and Likelihood** A scoring rule provides a local measure of fit or calibration given a predictive distribution and its corresponding labels. Specifically, for a datapoint $(x_i, y_i)$, let $\hat{y}_i$ be the predictive distribution of a model. Then, a scoring rule is any function that outputs a scalar evaluation of the goodness of fit: $S(y_i, \hat{y}_i)$. Such a scoring rule is called *proper* if $\mathbb{E}_{y \sim Q}[S(y, \hat{y})]$ is maximized when $\hat{y} = Q$, meaning that when the predictive distribution is perfectly calibrated and equal to the true label distribution, the score is maximal.

A common proper scoring rule is the log likelihood. Recall that for distributions over $[C]$, $p$ and $q$ that cross-entropy is $H(p, q) = -\sum_i p_i \log(q_i)$ and entropy is $H(p) = H(p, p) = -\sum_i p_i \log(p_i)$. In the classification setting the log likelihood is equivalent to the negative cross-entropy $S(y, \hat{y}) = -H(y, \hat{y}) = e_y^\top \log(\hat{y})$. Indeed, we see that for any predictive distribution $\hat{y}$ and label distribution $Q$, $\mathbb{E}_{y \sim Q}[S(y, \hat{y})] = -H(Q, \hat{y})$. If $\hat{y} = Q$ and our label distribution is uniform, then this is equal to the maximum entropy of $\log(|C|)$. When $\hat{y}$ is perfectly calibrated, the average negative cross-entropy will be equal to the negative entropy: $\mathbb{E}_{y \sim \hat{y}}[e_y^\top \log(\hat{y})] = \sum_i \hat{y}_i \log(\hat{y}_i) = -H(\hat{y})$. It is easy to see that this is a proper rule since for any predictive distribution $\hat{y}$ and label distribution $Q$, $\mathbb{E}_{y \sim Q}[S(y, Q)] - \mathbb{E}_{y \sim Q}[S(y, \hat{y})] = -H(Q) + H(Q, \hat{y}) = KL(Q || \hat{y}) \geq 0$, where the relative entropy or KL divergence is non-negative.

The definition of proper scoring rule works well when the label distribution is static and continuous, as when $Q$ is a point mass, the KL divergence becomes $H(Q, \hat{y})$, which is the case with hard labels. However, in most settings, $y$ is dependent on $x$ and $Q = \mathbb{E}_{(x,y) \sim \mu}[e_y | x]$ and $H(Q)$ is not known for each $x$. Furthermore, this gets even trickier when the predicted classes are non uniform, but under-represented classes require better calibration.

**Cross-Entropy vs Entropy** Due to the limitations of proper scoring rules, such calibrative measures are not meaningful measures of over-confidence on a per-datapoint level. Specifically, we consider the question of whether each datapoint is calibrated and note that the scoring rule of $H(y, \hat{y})$ does not give an inherent notion of calibration. Instead, we motivate the following definition of relative cross-entropy by noting that if $y \sim \hat{y}$, then $S(y, \hat{y}) - H(\hat{y})$ is a random variable with expectation 0.

### E.2 PROOFS OF LEMMAS IN § 4

*Proof of Lemma 1.* Let $i$ be the index corresponding to $y$. Then, by definitions of corresponding entropy measures,

$$RC(y, \hat{y}) = -\log(\hat{y}_i) - \left( \sum_j -\hat{y}_j \log(\hat{y}_j) \right)$$

$$= (\hat{y}_i - 1) \log(\hat{y}_i) - \left( \sum_{j \neq i} -\hat{y}_j \log(\hat{y}_j) \right) \tag{20}$$

$$\geq (\hat{y}_i - 1) \log(\hat{y}_i) + (1 - \hat{y}_i) \log\left( \frac{1 - \hat{y}_i}{|C| - 1} \right) \tag{21}$$

$$= (1 - \hat{y}_i) \left[ \log\left( \frac{1 - \hat{y}_i}{|C| - 1} \right) - \log(\hat{y}_i) \right]. \tag{22}$$

The third line uses the principle that entropy is maximized when uniform. To see this let $\alpha = \sum_{j \neq i} \hat{y}_j = 1 - \hat{y}_i$ and observe that the maximizing of the left hand side of the following equation admits the same argument as the subtrahend in Eq. 20,

$$\alpha^{-1} \sum_{j \neq i} -\hat{y}_j \left( \log(\hat{y}_j) - \log \alpha \right) = \sum_{j \neq i} -(\hat{y}_j / \alpha) \log(\hat{y}_j / \alpha),$$

and the right hand side is maximized with a uniform distribution. The lemma follows since both terms in Eq. 22 are positive. $\qquad \square$

*Proof of Lemma 2.* This follows since that $\hat{y}$ is a perfectly calibrated predictor, then

$$E_{y \sim \hat{y}}[RC(y, \hat{y})] = E_{y \sim \hat{y}}[C(y, \hat{y})] - H(\hat{y}) = 0.$$

$\qquad \square$

## F ADDITIONAL EXPERIMENTAL RESULTS

In this section, we expand upon the results that we presented in §5 and show additional experimental results and visualizations for model calibration. We start by presenting reliability diagrams for the heavy-tailed class distribution settings and continue with uncertainty and expected calibration error analyses.

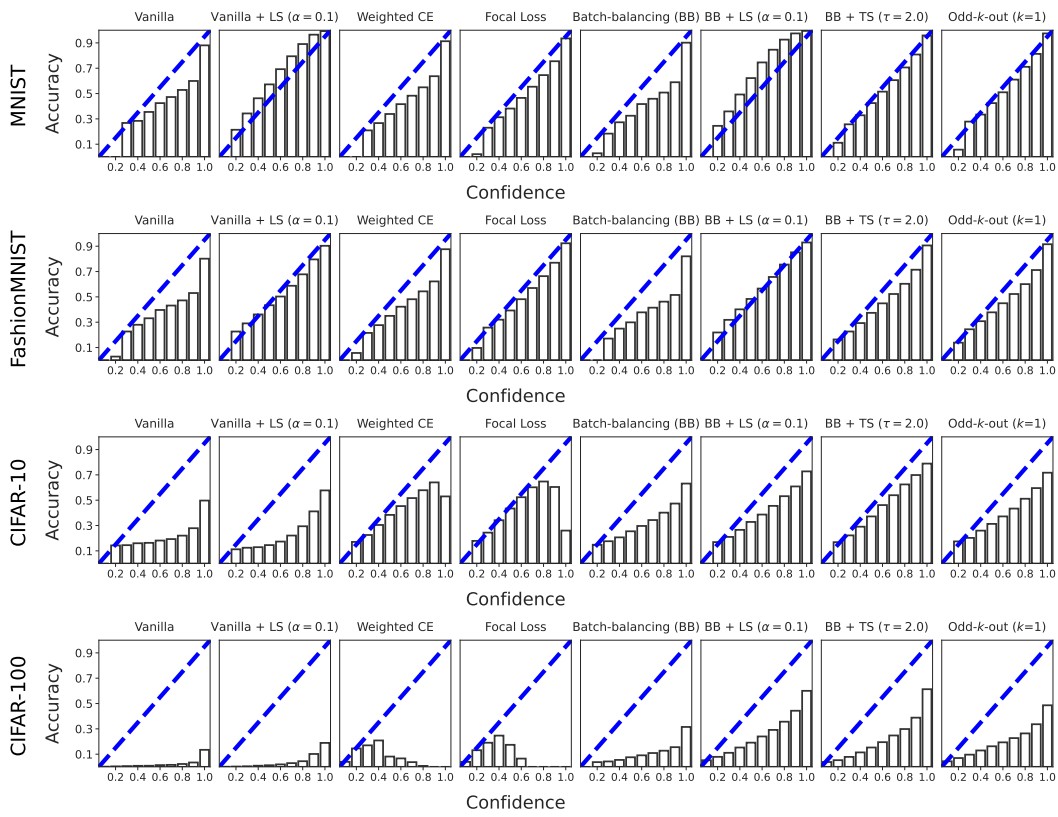

Figure 7: Reliability diagrams for heavy-tailed MNIST, FashionMNIST, CIFAR-10, and CIFAR-100. Confidence and accuracy scores are averaged over five random seeds and across the different number of training data points. Dashed diagonal lines indicate the best possible calibration.

## F.1 RELIABILITY

In Figure 7, we present reliability diagrams for heavy-tailed MNIST, FashionMNIST, CIFAR-10 and CIFAR-100. Both confidence and accuracy scores are averaged over five random seeds and across the number of training data points, similarly to Figure 3 in §5. Note that optimal calibration occurs along the diagonal of a reliability diagram, highlighted by the blue dashed lines. For training regimes with heavy-tailed class distributions, either OKO, batch-balancing in combination with label smoothing, or batch-balancing in combination with posthoc temperature scaling achieves the best calibration on the held-out test set.

## F.2 UNCERTAINTY

In Figure 8 we show the distribution of entropies of the predicted probability distributions for individual test data points across all heavy-tailed training settings for CIFAR-10 and CIFAR-100 respectively. In addition to the distribution of entropies of the predicted probability distribution for heavy-tailed training settings, in Figure 9 we show similar distributions for individual test data points across all balanced training settings for all four datasets. We find OKO to be very certain — $H(Q)$ is close to $\log(1)$ – for the majority of correct predictions and to be highly uncertain — $H(Q)$ is close to $\log(C)$ – for the majority of incorrect predictions across all datasets. Batch-balancing in combination with either label smoothing or temperature scaling shows a similar distribution of entropies for the incorrect predictions, but is often too uncertain for the correct predictions, indicating random guesses rather than certain predictions for a significant number of predictions (see Fig. 8).

## F.3 EXPECTED CALIBRATION ERROR (ECE)

Here we present ECE as a function of the number of data points used during training for both uniform and heavy-tailed class distributions for all four datasets considered in our analyses. We remark that for

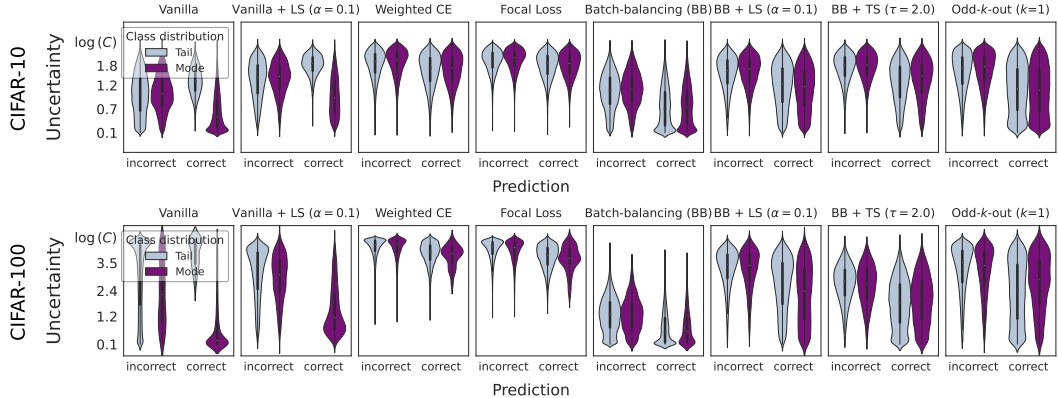

Figure 8: Here, we show the distribution of entropies of the predicted probability distributions for individual test data points across all training settings partitioned into correct and incorrect predictions respectively.

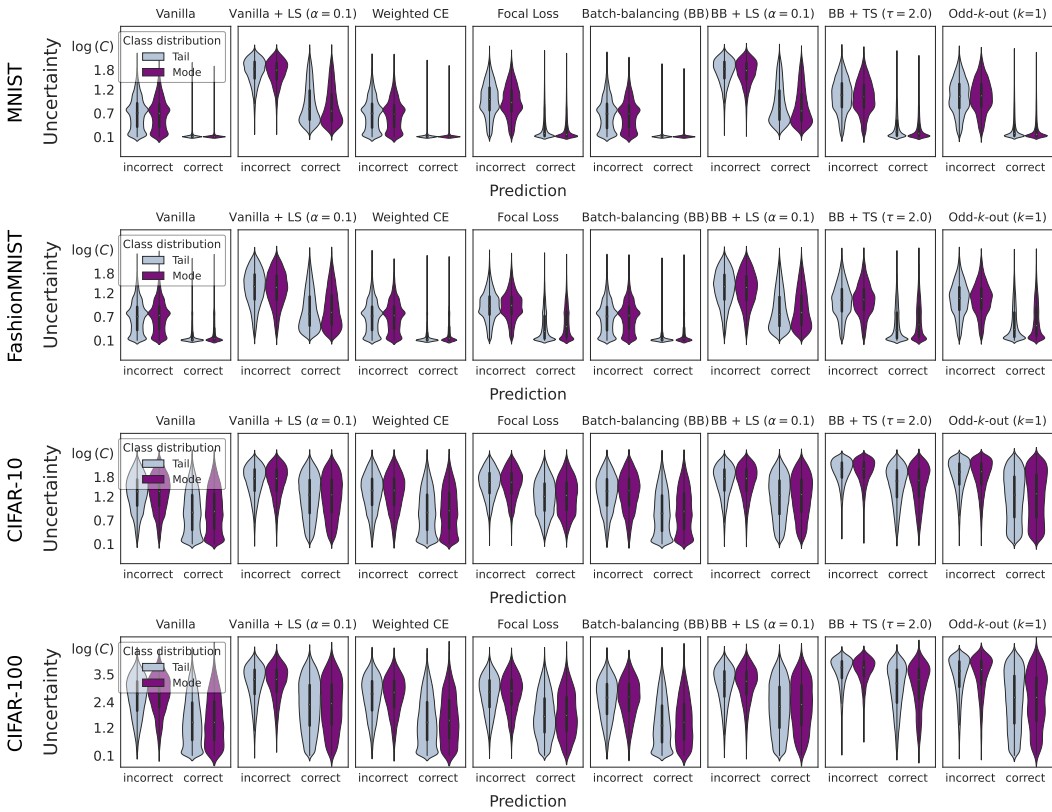

Figure 9: Here, we show the distribution of entropies of the predicted probability distributions for individual test data points across all training settings with a uniform class distribution partitioned into correct and incorrect predictions respectively.

every method the ECE was computed on the official test set. For balanced MNIST, FashionMNIST, and CIFAR-10 as well as for heavy-tailed MNIST OKO achieves a lower ECE than any other training method. For the other training settings, OKO is either on par with label smoothing or achieves a slightly larger ECE compared to label smoothing. This suggests that OKO is either better calibrated than or equally well-calibrated as label smoothing. The results are most striking in the low data settings.

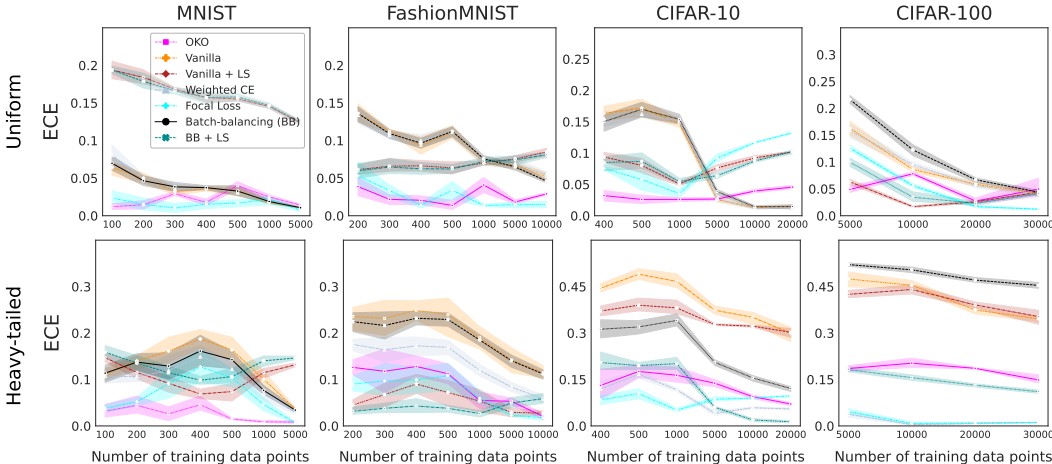

Figure 10: ECE as a function of different numbers of training data points. Error bands depict 95% CIs and are computed over five random seeds for all training settings and methods. Top: Uniform class distribution during training. Bottom: Heavy-tailed class distribution during training.

## F.4 How to select $k$, the number of odd classes

In this section, we compare different values of $k$ for generalization performance and calibration. Recall that $k$ determines the number of examples coming from the odd classes — the classes that are different from the pair class — in a set $\mathcal{S}$.

**Odd class examples are crucial**. Removing any odd class examples from the training sets — i.e., setting $k$ to zero — decreases generalization performance and worsens calibration across almost all training settings (see Fig. 11, Fig. 12, and Fig. 13). Although odd class examples are ignored in the training labels, they are crucial for OKO's superior classification and calibration performance. Odd class examples appear to be particularly important for heavy-tailed training settings.

**The value of $k$ does not really matter**. Concerning test set accuracy, we find that although odd class examples are crucial, OKO is fairly insensitive to the particular value of $k$ apart from balanced CIFAR-10 and CIFAR-100 where $k = 1$ achieves stronger generalization performance than larger values of $k$ (see Fig. 11). However, this may be due to the additional classification head that we used for predicting the odd class in a set rather than a special advantage of $k = 1$ over larger values of $k$.

We find larger values of $k$ to result in worse ECEs for training settings with a uniform class distribution and similarly low or slightly lower ECEs for heavy-tailed class distribution settings (see Fig. 12). Similarly, we find the mean absolute difference (MAE) between the average cross-entropy errors, $\bar{H}(P, Q)$, and the average entropies, $\bar{H}(Q)$, on the test sets for different numbers of training data points to be slightly lower for $k = 1$ than for larger values of $k$ for uniform class distribution settings and equally low or slightly larger for $k = 1$ compared to larger values of $k$ for heavy-tailed class distribution settings (see Fig. 13 for a visualization of this relationship and Tab. 6 for a quantification thereof). Across all training settings the MAE between $\bar{H}(P, Q)$ and $\bar{H}(Q)$ is the largest and therefore the worst for $k = 0$.

Table 5: MAE between entropies and cross-entropies averaged over the entire test set for different numbers of training data points. Lower is better and therefore bolded. This quantifies the relationships shown in Fig. 13.

| Training \ Distribution | MNIST uniform | MNIST heavy-tailed | FashionMNIST uniform | FashionMNIST heavy-tailed | CIFAR-10 uniform | CIFAR-10 heavy-tailed | CIFAR-100 uniform | CIFAR-100 heavy-tailed |
|---|---|---|---|---|---|---|---|---|
| Odd-$k$-out ($k$=0) / Pair | 0.091 | 0.318 | 0.136 | 0.542 | 0.657 | 1.370 | 2.088 | 3.739 |
| Odd-$k$-out ($k$=1) | **0.073** | **0.094** | **0.080** | 0.334 | **0.116** | 0.498 | **0.314** | 1.164 |
| Odd-$k$-out ($k$=2) | 0.224 | 0.123 | 0.204 | 0.208 | 0.352 | 0.194 | 0.527 | 1.040 |
| Odd-$k$-out ($k$=3) | 0.285 | 0.180 | 0.279 | **0.160** | 0.376 | **0.153** | 0.682 | **0.769** |

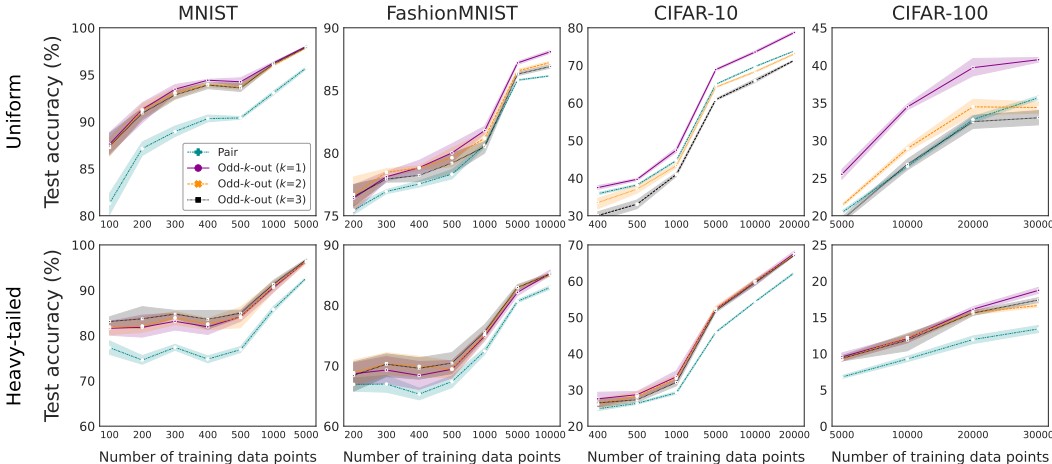

Figure 11: Test set accuracy in % as a function of different numbers of data points used during training. Error bands depict 95% CIs and are computed over five random seeds for all training settings and values of $k$. Top: Uniform class distribution during training. Bottom: Heavy-tailed class distribution.

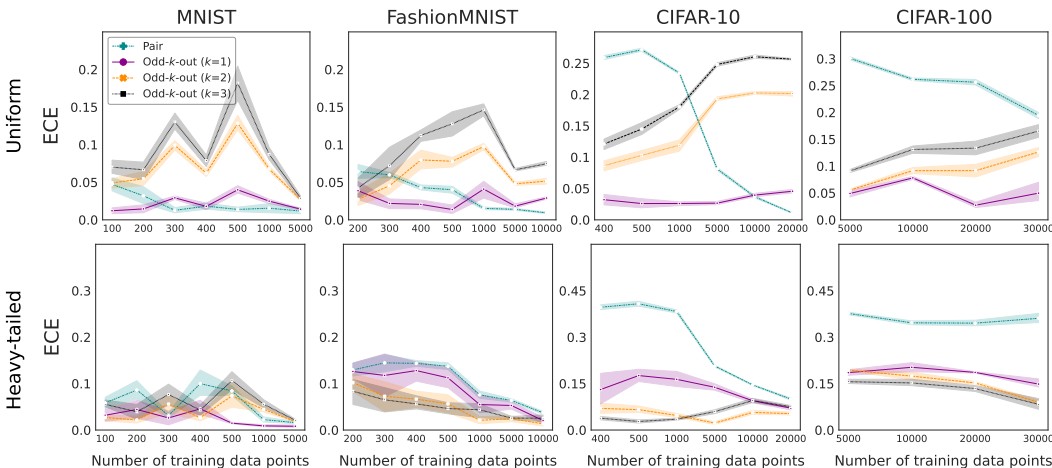

Figure 12: ECE as a function of different numbers of training data points. Error bands depict 95% CIs and are computed over five random seeds for all training settings and values of $k$. Top: Uniform class distribution during training. Bottom: Heavy-tailed class distribution during training.

## F.5 HARD VS. SOFT LOSS OPTIMIZATION

**Soft loss optimization spreads out probability mass almost uniformly**. Empirically, we find that the soft loss optimization produces model predictions that tend to be more uncertain compared to the predictions obtained from using the hard loss (see the large entropy values for models trained with the soft loss in Fig. 16). The soft loss optimization appears to result in output logits whose probability mass is spread out almost uniformly across classes and, thus, produces probabilistic outputs with high entropy values (often close to $\log C$). Fig. 16 visually depicts this phenomenon. These uniformly spread out probabilistic outputs lead to worse ECEs, where differences between the hard and soft loss optimization are more substantial for MNIST and FashionMNIST than for CIFAR-10 and CIFAR-100 respectively (see Fig. 15). Interestingly, for CIFAR-100 there is often barely any difference in the ECEs between the hard and soft loss optimization in the balanced class distribution setting, and in the heavy-tailed distribution setting, soft targets even yield lower ECEs than hard targets.

The test set accuracy of models trained with the soft loss is substantially worse in all balanced class distribution training settings (the differences appear to be more pronounced for MNIST and

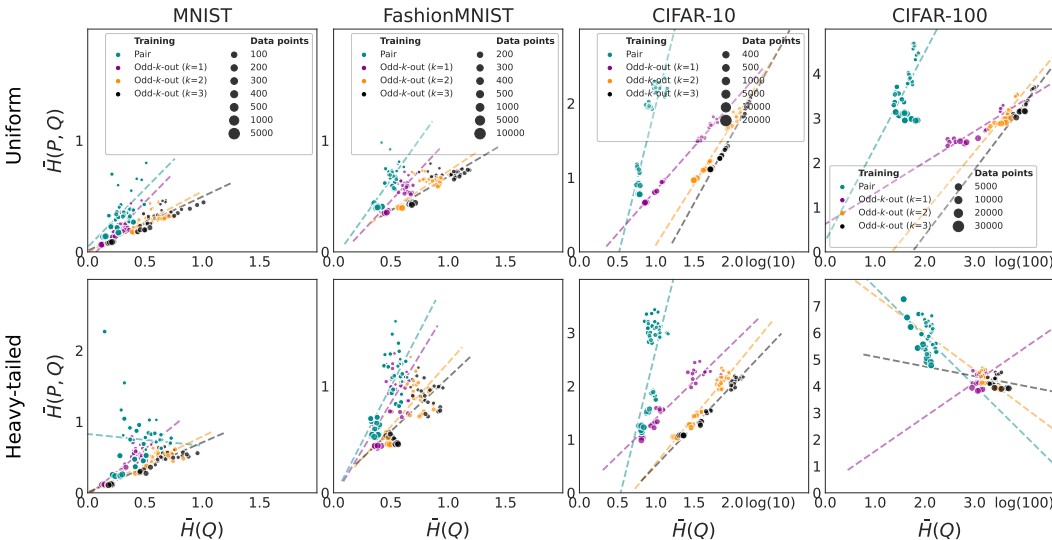

Figure 13: Average cross-entropy as a function of the average entropy for different numbers of training data points and different numbers of $k$. Top: Uniform class distribution during training. Bottom: Heavy-tailed class distribution during training.

CIFAR-100 than for FashionMNIST and CIFAR-10 respectively), but classification performance is only slightly worse compared to the hard loss in the heavy-tailed distribution settings (see Fig. 14).

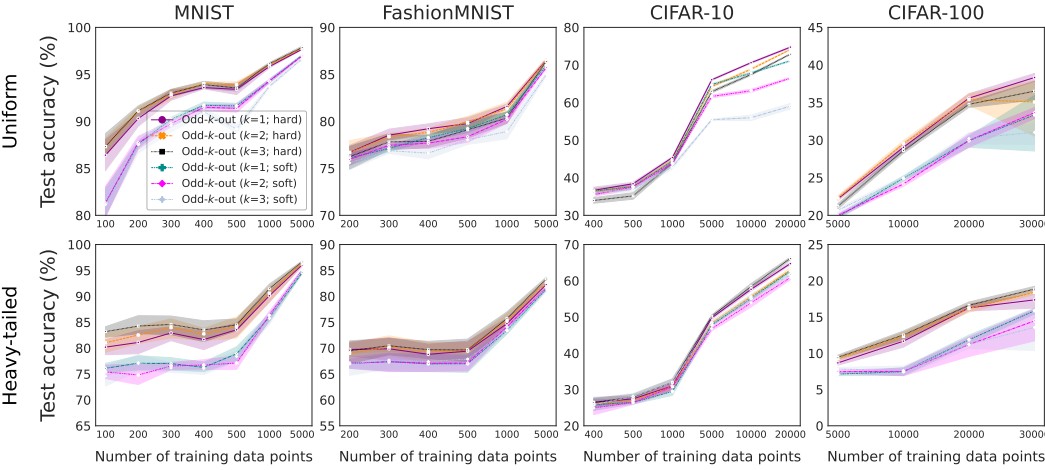

Figure 14: Test set accuracy in % as a function of different numbers of data points used during training. Error bands depict 95% CIs and are computed over five random seeds for all training settings. Top: Uniform class distribution during training. Bottom: Heavy-tailed class distribution.

Table 6: MAE between entropies and cross-entropies averaged over the entire test set for different numbers of training data points. Lower is better and therefore bolded. This quantifies the relationships shown in Fig. 16.

| Training \ Class distribution | MNIST | | FashionMNIST | | CIFAR-10 | | CIFAR-100 | |
|---|---|---|---|---|---|---|---|---|
| | uniform | heavy-tailed | uniform | heavy-tailed | uniform | heavy-tailed | uniform | heavy-tailed |
| Odd-$k$-out ($k$=1; hard) | **0.073** | **0.094** | **0.080** | 0.334 | **0.116** | 0.498 | **0.314** | 1.164 |
| Odd-$k$-out ($k$=2; hard) | 0.224 | 0.123 | 0.204 | 0.208 | 0.352 | 0.194 | 0.527 | 1.040 |
| Odd-$k$-out ($k$=3; hard) | 0.285 | 0.180 | 0.279 | **0.160** | 0.376 | **0.153** | 0.682 | 0.769 |
| Odd-$k$-out ($k$=1; soft) | 0.893 | 0.759 | 0.761 | 0.627 | 0.417 | 0.243 | 0.331 | 0.281 |
| Odd-$k$-out ($k$=2; soft) | 0.845 | 0.717 | 0.733 | 0.620 | 0.394 | 0.248 | 0.564 | **0.082** |
| Odd-$k$-out ($k$=3; soft) | 0.772 | 0.667 | 0.668 | 0.585 | 0.301 | 0.268 | 0.626 | 0.107 |

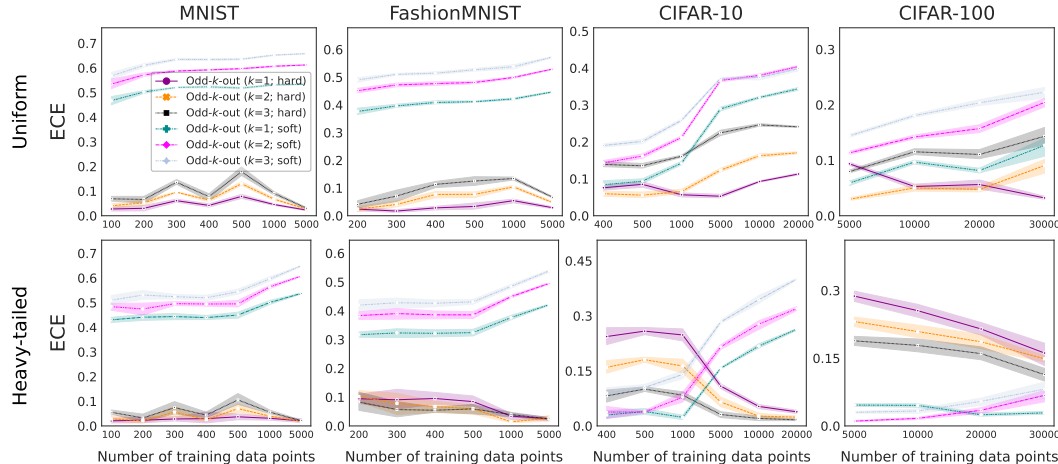

Figure 15: ECE as a function of different numbers of training data points. Error bands depict 95% CIs and are computed over five random seeds for all training settings. Top: Uniform class distribution during training. Bottom: Heavy-tailed class distribution during training.

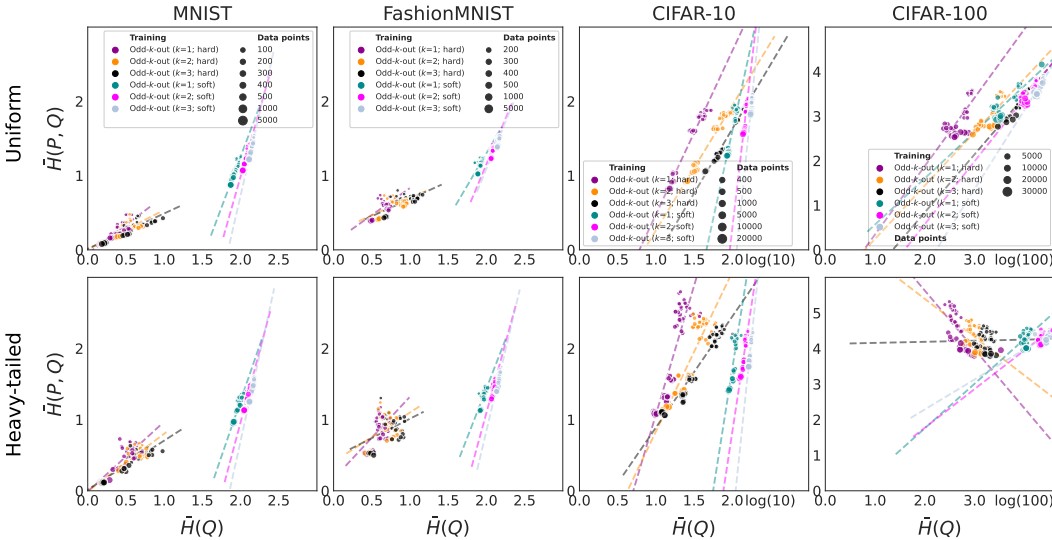

Figure 16: Average cross-entropy as a function of the average entropy for different numbers of training data points and different numbers of $k$. Top: Uniform class distribution during training. Bottom: Heavy-tailed class distribution during training.

### F.5.1 WHY DO SOFT TARGETS PRODUCE WORSE ACCURATE CLASSIFIERS?

Understanding why the soft loss optimization (see Eq. 7) results in worse accurate classifiers than the hard loss optimization (see Eq. 8) requires experiments that, unfortunately, go beyond the scope of this paper. However, using the theory that we developed for understanding the properties of OKO in combination with the experimental results from the ablation experiments for comparing the hard against the soft loss (see above), we can try providing an intuition about why the soft loss produces worse accurate classifiers. We remark that this should be read as an interpretation rather than a conclusion because we have no clear evidence for our intuition. Recall that the soft loss (see Eq. 7) in combination with Alg. 1 results in twice a mitigation of the overconfidence problem:

(a) The output logits are aggregated across all examples in a set and, thus, the loss is computed for a set of inputs rather than a single input (here, an input is an image). This step happens in both the hard and the soft loss optimization.

(b) Probability mass in the target distribution is spread out across the different classes in a set and, hence, transforms the majority class prediction problem — which is what the hard loss optimizes for — into a proportion estimation problem, which may make the model unnecessarily underconfident about the correct class.

Since the output logits aggregation step (a) is part of both the hard and the soft loss optimization, the hard loss may better strike the balance between mitigating the overconfidence problem and potentially shooting the optimization into local minima that amplify uncertainty where uncertainty/underconfidence is actually not desirable (which could be what is happening in the second step).

## G  COMPUTE

We used a compute time of approximately 50 hours on a single Nvidia A100 GPU with 40GB VRAM for all CIFAR-10 and CIFAR-100 experiments using a ResNet-18 or ResNet-34 respectively and approximately 100 CPU-hours of 2.90GHz Intel Xeon Gold 6326 CPUs for MNIST and FashionMNIST experiments using the custom convolutional neural network architecture. The computations were performed on a standard, large-scale academic SLURM cluster.

