# OpenReview forum: "Set Learning for Accurate and Calibrated Models"
_ICLR.cc/2024/Conference — ICLR 2024 poster_

### Official Review · Reviewer_ZKiJ · 2023-11-01

**Soundness:** 3 good
**Presentation:** 4 excellent
**Contribution:** 3 good
**Rating:** 8
**Confidence:** 4

**Summary:**

The paper proposes a new model training paradigm by minimizing the cross-entropy error for sets, rather than for individual examples. The method allows model to capture correlations across samples, and is believed to achieve better accuracy and calibration. The authors provide theoretical on calibration along with extensive experiment results to demonstrate the effectiveness of the method.

**Strengths:**

Originality: The set learning applied for training machine learning models is a pretty novel concept and paradigm change.

Quality: The paper is well written with high quality. The authors provide detailed motivations, illustrative figures and comprehensive appendix.

Clarity: The paper is written with clarity and easy to follow.

Significance: I believe the paper is of great significance to both academia and industry. Set learning is a new paradigm change to how people usually train ML models. The additional benefits brought upon on calibration is important for various applications in industry.

**Weaknesses:**

Mainly I have some questions on the experiment.

1. The authors set the maximum number of randomly sampled sets to the total number of training data points, such that the gradients updates remain the same. However, does this effectively make the proposed method sees k multiple more data points compared to the standard setting? It would be interesting to see how the experiment results look like when two algorithms are of the sample complexity.

2. Is there any intuitions behind why the hard loss always outperform the soft loss?

**Questions:**

See weaknesses.

---

> ### Author Response · Authors · 2023-11-17
> **Author response**
>
> Thank you very much for your positive review! We appreciate it and agree that substantial benefits for calibration could be of great significance to both academia and industry. Hence, we are glad to see that you think likewise. Here are our responses to your questions:
>
> 1.) We agree in principle that this would be interesting, and when performing the experiments in the paper we did indeed account for this: To determine the maximal number of update steps to use in our experiments, we performed an extensive grid search using the *vanilla* baseline. We then used that number for all experiments in our paper. We additionally used an *early stopping* training criterion, meaning we stopped training when the training process converged (no more improvements on a held-out validation set). Thus, all methods converged before hitting the maximal number of iterations. This means that increasing the number of update steps would not change the outcome (at least not substantially).
>
> 2.) We ran a full battery of ablation experiments for comparing hard and soft losses on our various training settings. We elaborate on this more thoroughly (and report our findings) in the **general response** to all reviewers (see above). We will add results, plots, and an additional section on the intuition behind *soft vs. hard loss* differences to the Appendix of our manuscript early next week!

---

> ### Author Response · Authors · 2023-11-20
> **Hard vs. soft loss optimization**
>
> We ran additional ablation experiments to compare the hard and the soft loss optimizations against each other. We added plots showing the results from these experiments to a new section F.5 in the Appendix. We compare test set accuracy and calibration performance across all training settings (class-balanced and heavy-tailed) and for all datasets (our findings are outlined in the comment above). In addition, we added a subsection as part of this new section on "Why soft loss optimization produces less accurate classifiers?" to provide a well-informed intuition about why the soft loss produces worse classifiers than the hard loss. This subsection can be found in F.5.1. See the **general response** for more details.

---

### Official Review · Reviewer_S9eW · 2023-11-06

**Soundness:** 2 fair
**Presentation:** 2 fair
**Contribution:** 2 fair
**Rating:** 6
**Confidence:** 3

**Summary:**

The submission addresses the problem of multi-class learning, where the purpose is to reduce the overconfidence of the model by using a specific decomposition of the problem called odd-k-out learning. OKO learning amounts to separate a positive class (or pair class) in which two instances are randomly selected, from a set of negative classes (or odd class) for each of which one instance is selected. The authors show that this makes it possible to obtain less certain model outputs in low-density regions, and thus to restrain miscalibration. Some properties of OKO are rapidly presented, before experiments are reported and a conclusion is drawn.

**Strengths:**

The language is overall fine.

The experimental results seem convincing.

**Weaknesses:**

OKO is only rapidly presented, through how instances are constructed. The overall procedure remains unclear.

The properties of OKO are succinctly mentioned, but the theoretical reasons for which OKO outperforms classical classifiers are not crystal clear.

The case considered in the experiments seems to be a very special case, and it is difficult to see whether the conclusions drawn can be generalized.

**Questions:**

The "set learning" problem you mention in the introduction is not formally defined, nor is the odd-one-out scheme in Section 2.

As far as I understand, OKO consists in transforming the classification problem into a binary classification problem, as made in the "error-correcting output coding" decomposition strategy. Could you explain the difference between OKO and ECOC ?

In Section 3, you explain how to construct a training example for the OKO setting. I assume this operation is repeated. Could you elucidate how instance generation can be repeated in the global OKO setting ?

Could you provide information on the results obtained using the soft loss, and/or insights regarding why the hard loss would give better results than the soft loss ?

Section 4 is very short. Theorem 1 seems to be limited to the example discussed. The section seems to call for a discussion on the different natures of uncertainty (i.e., aleatory vs epistemic uncertainty) and on the different treatments to be applied accordingly.

You emphasize the similarity between the relative cross-entropy for calibration you use and the KL divergence. Can you elucidate the advantages of RCE for calibration over KL divergence ?

You set $k=1$ for all experiments: this seems to correspond to a very special case of OKO. Could you explain the impact of this choice ? This somehow mitigates the conclusions that can be drawn from the experimental part, as it seems difficult to generalize them. Can you elaborate ?

Typos and writing :
- I do not understand the notation in Equation (3) ("$f_\epsilon \in ...");
- "the occur infrequently" (page 5 bottom);
- "can be boosted by predicting the odd class using an additional classification head" (page 6) : I do not understand this sentence.

---

> ### Author Response · Authors · 2023-11-18
> **Author response**
>
> Thank you for your thorough review! We appreciate that you took the time to give us constructive feedback.
>
> Regarding the weaknesses:
>
> - We will improve the presentation of our contributions in the next update of the manuscript (which we will post early next week). See point two in the **general response** for how we plan to do this.
>
>
> - The idea behind OKO is to identify the majority class of a set of inputs. Here we focus on the special case where the sets consist of 2 examples from the majority class and 1 example from the minority ($k=1$). We demonstrated empirically that this setting generalizes to other settings. For space reasons we had to move these experiments to Appendix F.4. That section shows that the conclusions generalize (the particular value of $k$ rarely matters but setting $k>0$ is crucial.). We thus decided to focus on $k=1$ in our main paper because we think that this is the setting that is most relevant to practitioners.
>
> For your specific questions:
>
> > The "set learning" problem you mention in the introduction is not formally defined, nor is the odd-one-out scheme in Section 2.
>
> In these sections, we avoided a more formal introduction specifically because we also want to talk about related work, which does not necessarily share the same formalisms as we do. Rather, we want to talk broadly about other methods that use set learning. A very formal introduction (needed for the proofs) can be found in the Appendix.
>
> > As far as I understand, OKO consists in transforming the classification problem into a binary classification problem, as made in the "error-correcting output coding" decomposition strategy. Could you explain the difference between OKO and ECOC?
>
> This is a very interesting question. We had not seen the connection to ECOC, thank you for pointing it out! We will include it into our related works discussion. The idea behind ECOC is to look at multiclass classification by training several different binary classifiers. In contrast, with OKO one trains just a single final classification model (not several), and that model still makes a multiclass prediction (in ECOC, every model makes a binary decision). OKO is also focused on delivering well-calibrated models, which AFAIK is not a main goal of ECOC.
>
> > In Section 3, you explain how to construct a training example for the OKO setting. I assume this operation is repeated. Could you elucidate how instance generation can be repeated in the global OKO setting ?
>
> Algorithm 1 shows how to construct a single training example for OKO. By running this algorithm repeatedly (for multiple steps), one can construct a whole dataset of sets --- which we refer to as $\mathcal{D}^{\prime}$ --- for using OKO.
>
> > Could you provide information on the results obtained using the soft loss, and/or insights regarding why the hard loss would give better results than the soft loss?
>
> Please see our **general response** where we provide an answer to this. We will add a new section in a revised version of the manuscript.
>
> > Section 4 is very short. Theorem 1 seems to be limited to the example discussed. The section seems to call for a discussion on the different natures of uncertainty (i.e., aleatory vs epistemic uncertainty) and on the different treatments to be applied accordingly.
>
> Theorem 1 is just meant to give some intuition about how OKO behaves in a particular setting and help explain why it may exhibit good calibration. It indicates that, for the setting in Theorem 1, input regions with fewer samples, which may be interpreted as a higher degree of epistemic uncertainty, manifest as aleatoric uncertainty in the OKO test time outputs. While it is conceivable that such regions indeed have low aleatoric uncertainty, and that the few samples do indeed characterize the entire region, in safety-critical applications it is often desirable for the network to err towards uncertainty. We will include this point with an explanation of *aleatoric* and *epistemic* uncertainty in the next draft.
>
> > You emphasize the similarity between the relative cross-entropy for calibration you use and the KL divergence. Can you elucidate the advantages of RCE for calibration over KL divergence?
>
> These questions were raised by several reviewers. Therefore, we discuss it in the **general response**.
>
> > You set k = 1 for all experiments: this seems to correspond to a very special case of OKO. Could you explain the impact of this choice? This mitigates the conclusions that can be drawn from the experimental part, as it seems difficult to generalize them. Can you elaborate?
>
> We did not have space to include detailed results about this in the main text, but Appendix Section F.4 answers this question. The section shows through empirical evidence that this special case generalizes well to more general cases because the particular choice of $k$ does not matter but $k>0$ is crucial.
>
> > Typos and writing:
>
> Thank you for pointing these out. We will fix these and upload a new version asap! :)

---

> > ### Comment · Reviewer_S9eW · 2023-11-21
> > **Follow-up**
> >
> > I'd like to thank the authors for their answers to my questions and comments, which shed some light on their work.
> >
> > I'll raise my score accordingly.

---

> ### Author Response · Authors · 2023-11-21
>
> Thank you for your response and for raising your score!

---

### Official Review · Reviewer_ALEr · 2023-11-07

**Soundness:** 2 fair
**Presentation:** 1 poor
**Contribution:** 2 fair
**Rating:** 3
**Confidence:** 5

**Summary:**

This paper proposed a modified loss function that can achieve more accurate and more calibrated classification models. The idea is to define a loss based on a set of instances (and their ground truth). Within each set, two instances are from the same class, and the other k instances are from k distinct classes. The proposed soft loss modifies the cross-entropy loss by replacing the instance-level regression function f by the sum of f functions over the set, and the instance level one-hot ground-truth group indicator by the sample proportion in the set. For the hard loss variant, the instance level one-hot ground-truth group indicator is replaced by the indicator of the most common class in the set.

The authors contend that such modifications foster improved accuracy and calibration, a claim substantiated through comprehensive empirical studies. Nonetheless, the paper does not thoroughly explain the underlying benefits of this proposed method, leaving the reader with an incomplete understanding.

Additionally, the paper presents a novel calibration metric dubbed 'relative cross-entropy' (RC). This introduction, however, appears to be executed hastily, with a significant number of details absent, which might leave the concept open to further exploration and clarification.

**Strengths:**

The proposed loss function seems to be interesting and numerical results show good performance.

**Weaknesses:**

The theoretical underpinnings presented in the paper are not robustly developed.

The organization of the manuscript lacks coherence, with various concepts introduced but not adequately interconnected or explicated.

For further elaboration on these points, please refer to the questions outlined below.

**Questions:**

1. page 1 stated that "By construction, this paradigm ignores information that may be found in correlations between sets of data." I do not think "correlation" is the correct word here. I do not see how correlation is explicitly exploited in the newly defined loss function.  Moreover, shouldn't it be the relationship among points in the same set instead of the relationship among different sets?

2. Page 3, algorithm 1: Is it guaranteed that each class is sampled at least once as the pair class? If not, then some class may not be represented in the model.

3. Page 3, equation (1). The inner product is a summation over classes, and the loss function will be summed over sets. Hence, the proposed soft loss essentially boils down to an exchange of the order of summations: the traditional cross-entropy loss sum over instances (in a set) first, and then over the classes; in the proposed loss, one sums over the classes first, and then over the instances within a set. It is then intriguing as to why the improved performance? In a sense, (2) is even closer to the cross-entropy loss than (1), since in (2) only the instance level f function is replaced by the aggregate of (k+2) f functions. In the numerical studies, the soft loss is not presented at all because "we found the hard loss to always outperform the soft loss". I cannot help wondering if the soft loss is even better or on par with the baselines?

4. Since k does not really matter, can it be zero? In fact, can we have more than 2 instances from the dominating class in a set. In this case, (2) would be very similar to a multi-instance learning method.

5. Page 5 stated that "The key observation from Theorem 1 is that, although x = 1 or x = 2 have zero entropy and are thus low-entropy regions, OKO is still uncertain about these points because the occur infrequently". However, this is not necessarily a good thing (or not good enough). For example, the result of Theorem 1 does not say that the calibration is correct. In fact, I am not even sure how to define the calibration in this example because the limit portability distribution seems weird to me: it is distributed as 1/0 when epsilon is greater than 0 and 0/0 when epsilon = 0. Specifically, the limit of epsilon = 0 is a singular case. Can this be generalized to more general settings? In summary, Theorem 1 is about a fairly special case, and even in that case it does not explicitly show that the calibration is correct.

6. I am very puzzled by the subsection titled "Relative Cross-Entropy for Calibration" in Section 4. This RC has nothing to do with OKO and should not be part of a section titled "Properties of OKO". Moreover, this subsection reads very out of context. It is unfortunate that relevant discussion is in the appendix, making it very difficult for the reader to see its relevancy to the topic being discussed in the main text.

7. Page 6 stated that "RC is no longer proper due to this zero mean." What does "proper" mean?

8. Page 6 stated that "we report results for a version of OKO with k = 1 where in addition to the pair class prediction (see Eq. 2) a model is
trained to classify the odd class with a second classification head that is discarded at inference time." How is the overall objective including the second classification head defined? A mathematic formula and/or a graph of the network structure would be helpful.

9. Page 6 stated that "For simplicity and fairness of comparing against single example methods, we set the maximum number of randomly sampled sets to the total number of training data points ntrain in every setting." In my opinion, it is still not a fair comparison since for the OKO method there are 3n instances out of the n sets.

10. It is unclear to me what I should look for in Figure 4? Why is one method better than another and how is such an advantage shown in the figure? More specific pointers should be helpful.

11. Page 9 stated that "OKO is a theoretically grounded learning algorithm that modifies the training objective into a classification problem for sets of data. We show various consistency proofs and theoretical analyses proving that OKO yields smoother logits than standard cross-entropy, corroborated by empirical results." I strongly disagree with this statement. In the main text there is only one theorem, which, as I mentioned above in question 5, is not strong enough. There is no consistency results in the main text.

12. In C.1 it stated "Before proving Proposition 2 from the main text ...."  Proposition 2 is in the appendix, not in the main text.

13. It is clear to me that this paper was writing as a long, comprehensive paper. However, it was trimmed down to 9 pages in order to submit to ICLR, in a rush. For this reason, the presentation of the topics is very poor. There are too many topics in this 9 page paper, and the author did not make a convincing case advocating for the proposed method. Moreover, because of the careless reduction of topics, there are topics that read out of context (e.g. the introduction of RC), and there are topics that should have been explained more (e.g. theoretical justification of the proposed loss.) I think it may serve the readers better by submitting a comprehensive version of the paper to a journal such as JMLR, instead of rushing through a conference.

---

> ### Author Response · Authors · 2023-11-17
> **Author response (1/2)**
>
> Thank you for your feedback! Below are our responses to all of your 13 questions:
>
> 1.) Maybe “shared latent information” might indeed be a better word for what we’re trying to express. We think OKO is better able to tease out such information because it allows the model to compare data points (both from the same set as well as among different sets), so that every gradient descent step is influenced by more than just a single data point. Judging by our empirical results, we think it’s fair to say that our model, indeed, appears to learn more sensible decision boundaries wrt. Calibration.
>
> 2.) The pair class is chosen uniformly at random from the total number of classes. So the probability of each class being drawn goes to one as the number of samples goes to infinity. In more practical terms: we are confident that all classes are drawn equally often as pair classes.
>
> 3.) This is an interesting observation! We now give an exact comparison of hard vs. soft loss in the Common Responses to all reviewers. We included plots and analyses in the Appendix of our manuscript.
>
>
> 4.) The connection to Multi-Instance Learning (MIL) for $k=0$ is reasonable. Thank you for pointing it out! We agree that one can have more than $2$ instances from the majority class. Appendix Section F.4 contains ablation for various settings of $k$, including $k=0$. As stated in the main text, the particular choice of $k$ does not matter much but $k > 0$ is substantially better than $k=0$ across all training settings (see Section F.4 in the Appendix). In preliminary experiments, we tried larger numbers of images for the majority class when $k=0$ and did not observe substantial differences between 2, 3, and 4 images in the majority class when $k=0$. Hence, we conclude that $k > 0$ is important but not its particular value. Of course, one could try different combinations of the number of images in the majority class and $k$ but this is an exponentially large combinatorial problem and we leave this to future analyses.
>
>
> 5.) The desirability of the phenomenon highlighted in Theorem 1 is dependent on the application. It is most desirable when one would like to err towards uncertainty for regions of the input space that are sparsely sampled. This might be beneficial for a safety-critical setting where it would be better for the network to indicate less certainty in sample regions with little data rather than having the network assign high certainty from just a sample or two. We can include this point in the next draft, if you want.
>
>
> 6.) We will connect RC better with OKO and how RC helps understand the empirical fact that OKO produces greater entropy predictions by showing that the cross entropy vs entropy diff is more centered at zero (see our general response).
>
>
> 7.) The definition of proper was, unfortunately, moved to the Appendix and we will also move that sentence as well. A proper score is essentially a score that is maximized in expectation when labels are from the same distribution as the calibrating distribution. However, since RC has an expectation of 0 on the same distribution, and it can be positive, or negative, it is in fact not a proper scoring rule.
>
>
> 8.) The objective of the 2nd head is cross entropy between the odd class label $(y_{i})$ and the logits of the odd class representation $(x_{i})$. In other words, the full loss function in this case was $L_{\text{oko}}^{\text{hard}}(S_y, f_\theta (S_x)) + y_i \log \theta(x_i) $. We will clarify this in the next version of the manuscript.

---

> ### Author Response · Authors · 2023-11-17
> **Author response (2/2)**
>
> 9.) We can see your point and where you are coming from, but we would still like to disagree here: It is indeed true that on average, OKO converges sooner, because it processes more information per update step. However, we made sure that every method could run until convergence, so we are very confident that more examples would not have helped the baselines improve their performance. Also, recall that $n$ does not change. So, if you train each method until convergence, it does not matter how often a model has seen the same image.
>
>
> 10.) In Figure 4, we show the distribution of entropies of the predicted probability distributions for individual test data points (i.e., at inference time) partitioned into correct and incorrect predictions respectively. The $x$-axis represents correct vs. incorrect predictions (hence, its label is binary), and the $y$-axis represents the entropy, or uncertainty if you will, of the model’s softmax outputs, and therefore its label is continuous. The range of the entropies goes from $\log(1) = 0$ to $\log(C)$ where $C$ is the number of classes in the data.
>
>
> 11.) Thank you for this feedback. After seeing your and the other reviewer’s reaction, we agree that our claims were too strong in this regard. Please see the **general response** above for more details about how we intend to fix this.
>
> 12.) Thank you for pointing this out. We will fix this accordingly.
>
>
> 13.) Thank you for the blunt feedback, we appreciate the honesty. As we addressed in the general response to the reviewers, we agree that our claims about the theoretical contributions were too strong. We toned them down, and improved the presentation of our results, and the general flow of our text.  We would kindly ask you to take a look at the revised version of the PDF that we will upload early next week: You seem to agree that our approach is interesting and our empirical results look good. So, we would kindly ask you to consider whether the results we present here are truly something the ICLR community is not interested in.
>
>
> We hope that this answers your questions! Let us know if there’s anything else you’d like to know or that we can do to increase your rating.

---

> > ### Author Response · Authors · 2023-11-23
> > **Reminder**
> >
> > Dear Reviewer ALEr,
> >
> > we would appreciate if you could respond to our rebuttal and let us know if it has addressed your concerns. Is there anything else that we can do?
> >
> > The Authors

---

### Official Review · Reviewer_se1z · 2023-11-08

**Soundness:** 3 good
**Presentation:** 2 fair
**Contribution:** 3 good
**Rating:** 8
**Confidence:** 3

**Summary:**

This paper introduces a new training method, which the authors call ``odd-k-out'' for training machine learning classifiers, in order to reduce the problem of over-confidence observed in large models trained on relatively small datasets (which often end up interpolating the training set).

The method essentially boils down to sampling uniformly at random two distinct classes $k_1, k_2$, and then two uniformly random examples $x_1, x_2$ from the first class ($y_1, y_2 = k_1$), and one example $x_3$ from the second class (with $y_3 = k_2$), and taking a gradient step in the direction minimizing the loss given by $e_{k_1}^T \log(\mathrm{softmax}(\sum_{i \in \{1,2,3\}} f_{\theta}(x_i)))$.

The authors report improvement in terms of accuracy and importantly calibration error with respect to a number of benchmarks, and show that their method provides an approach to handle pressing issue stemming from the fact that out-of-the-box machine learning models tend to preform poorly on sub-populations that are underrepresented in the training set. They also provide a theoretical insight into how this assertion can be justified in a simplified model (Theorem 1) --- essentially saying that in specific situations, if on vanishingly small subsets of the universe the features $x_i$ determine the outcome exactly, the OKO-minimizers will only put confidence $2/3$ of this specific outcome (on aforementioned small subsets).

The introduced method seems to be fairly close to ``batch balancing + label smoothing'' --- the authors discuss briefly this method in paper and use it as a comparison point. Batch balancing here refers to sampling examples by first sampling a uniform class, and then sampling a uniform example from the class (as opposed to just sampling uniform example from the entire dataset), which is a standard method to remedy class imbalance in the training set. Label smoothing corresponds to minimizing loss $\ell(f_\theta, \tilde{y}_i)$ where the desired label $\tilde{y}_i$ has some fraction $\alpha$ of mass on the actual class $y_i$ and $1-\alpha$ fraction of mass evenly distributed across remaining classes --- therefore addressing the problem of over-confidence.

In the OKO method, the load balancing is done explicitly, whereas the label smoothing is somewhat implicit --- while looking at a gradient of an expression $e_{k_1}^T \log(\mathrm{softmax}(\sum_{i \in \{1,2,3\}} f_{\theta}(x_i)))$ which is being minimized, we see that what happens is essentially adding some linear combinations of gradient pushing the outcome of the classifier $f_{\theta}$ to be more confident of outcome $y_1$ on all three examples $x_i$ --- of which two has actual label $y_1$, and the third has uniformly different label --- this feels fairly close to just taking a random class $k_i$, a random example $x_i$ and shifting the weights towards higher confidence of class $k_i$ with probability $2/3$, vs higher confidence for a different (uniformly distributed) class with probability $1/3$ --- essentially label smoothing.

In contrast with label smoothing, as authors show with their Proposition 3, that in a toy example in which labels are directly determined by the feature (i.e. $x_i \in [C]$ is uniformly random, and the corresponding label is $y_i = x_i$, then minimizing OKO-objective (over the class of all functions from $[C]$ to the probability simplex $\Delta([C])$ leads (as it is desirable) to diverging logits: i.e. the predictions $\tilde{f}(x)$ converge to $(0, \ldots, 1, \ldots 0)$ with the label $1$ on the correct label $x$.

**Strengths:**

The paper discusses quite extensively the existing literature in the topic, deals with extremely pressing issue in the machine learning community, providing a new method to address the problem. As such it has potential of having significant impact, and the paper provides experimental evidence that their proposed method outperforms the known results in terms of calibration and accuracy on standard benchmarks for.

**Weaknesses:**

Given how close the newly proposed method is to label smoothing with batch balancing in essence, it might be worthwhile to discuss in much more details how the proposed method is in fact different than label smoothing, especially in main body of the paper. Highlighting a bit more the Proposition 3 which they prove in appendix could improve interpretability of their paper.

On page two, in the paragraph "Empirical", the authors write "OKO is a principled approach that changes the learning objective by presenting a model with _sets of examples_ instead of individual examples, as calibration is inherently a metric of sets". This statement seems to run into a fallacy. Indeed, calibration is a metric of sets (or, one can say, distributions over pairs of predictions and outcomes, usually given as a uniform distribution on a finite set),  and indeed the learning objective here is given by presenting a model with _sets of examples_. But those two sets have nothing to do with each other. The easiest way to see it, is that the authors suggest using as a learning objective sets of size $3$ (two in-class examples, and one out-of-class (see ``Experimental details`` page 3) --- and on a set of size $3$ any statement about calibration (even for binary classification) is meaningless --- one needs at least a couple dozens of examples to this.

The main theoretical result is Theorem 1: it considers the following toy scenario, with two classes, where the feature space is $\{0, 1, 2\}$, and the population distribution $F_\varepsilon$ is given as follows. With probability $1-\varepsilon$ we have $x_i = 0$ and $y_i$ is uniformly random class $\{0, 1\}$. With probability $\varepsilon/2$ we have $x_i=1, y_i=0$ and with probability $\varepsilon/2$ we have $x_i=2, y_i=1$. They show that as $\varepsilon \to 0$, the OKO-minimizers tend to something that on $x=1$ outputs the predictions $(2/3, 1/3)$. This is presented as evidence that the OKO, as desired, is not over-confident in regions that are severely underrepresented in the distribution.

I can see two issues with this argument: first of all it is not immediately clear that this behavior is desirable at all, it would be great to provide more of a discussion when it is the case. Even if relatively small fraction of a population distribution has a specific feature that turns out to determine exactly the outcome, as long as this is in fact a feature of population distribution (and not an artifact of small _number_ of examples in the training distribution), it is absolutely reasonable to report this outcome with high confidence on new example that exhibit this feature. It doesn't seem to be too far-fetched either: one can easily imagine having 1% of both population distribution and a training set exhibiting a feature that determines the outcome $y_i$ --- if trained on training sets with several hundred millions of examples, this leads to millions of examples with the aforementioned feature. In a situation like this it is not only reasonable, but in fact desirable to report the determined outcome with high confidence (and it is not difficult to come up with scenarios where this could be crucial).

Second issue is what the authors do not discuss, which is the fact that this behavior proved in their theorem is not an effect of the classes $x_i=1$ and $x_i=2$ represented by small fraction of the population distribution at all --- contrary to the discussion in front of the theorem. In fact, back-of-the-envelope calculation suggest that even if probabilities for $x_i=0, x_i=1, x_i=2$ where all equal $(1/3)$, and $x_i=1$ determined $y_i=0$, $x_i=2$ determined $y_i=1$, (just as in their setup), just as the label-smoothing, would learn to classify examples with $x_i=1$ as having probability of $y_i=0$ to be some number distinctively separated from $1$ (regardless of the amount of training data). This might be even more undesirable in specific situations, and crucially this sheds a different light on how to interpret their Theorem 1: the effect they discuss (of producing a confidence for the label strictly bounded away from $1$) is not introduced by the fact that only vanishingly small fraction of the population has features that seems to determine the label $y_i$ (as suggested by the name of the paragraph ``OKO is less certain in low-data regions''), it is in fact just a consequence of an implicit label-smoothing.

Minor issues regarding the presentation:
There is a lot of plots, stacked together in extremely small space, often on each of the multiple sub-figures in a figure, there is 8 different plots with trend lines, making most of them completely unreadable, and as such not adding much value. That is particularly true about Figure 1, Figure 3, Figure 4, Figure 5.

The statement of Theorem 1 is somewhat awkward: it consist of two separate sentences --- the first part states using symbols, that for any $\varepsilon$ there exist a minimize for the OKO objective on a distribution $F_\varepsilon$ --- since a statements of this form are often trivial (and the nature of local/global minimizes is the object of study), it takes a bit of time for a reader to understand its importance in this specific situation. It might be worth to expand it slightly --- as it is, the ``Furthermore, ...'' part of the theorem seems on the first glance as the only one conveying meaning.

I am rather confused by the notion of "relative cross-entropy". This seems to be just KL-divergence. The authors say that it is ``very similar yo KL divergence but with different entropy term''. The $RC(P, Q)$ is defined as $H(P, Q) - H(Q)$ (Definition 1), the KL-divergence is $D_{KL}(P || Q) = H(P, Q) - H(P)$, hence $RC(P, Q) = D_{KL}(Q || P)$. The authors say that it is (unlike KL-divergence) not always non-negative, which doesn't seem possible.

The statement of Lemma 1 does not seem to match the explanation  preceding it, and it is not clear what it was meant. Anyway, the conclusion seems true, since the KL-divergence is indeed non-negative without any additional assumptions.

**Questions:**

Most of my questions was implicitly stated in the discussion about weaknesses. I find the Theorem 1 fairly unconvincing as a desired property of the proposed training method --- it seems like informative Theorem highlighting \emph{a property} of OKO, not necessarily an argument for using it. In light of superficial similarities with batch balancing + label smoothing, expanding a bit on similarities differences between those (and particularly scenarios where OKO seems to behaving closer to what we would expect than label smoothing) would be nice. The Proposition 3 is a great step in this direction, so potentially just highlighting it and providing more detailed discussion about it in the main body of the paper would be nice.

The Proposition 2 is said to be interpreted as "OKO directly encouraging the risk not to overfit" -- it is unclear to me whether this interpretation is justified. It is not clear what it means that when restricting all but one entry in the purported matrix of logits $F_{i,j}$ the OKO objective have a global minimizer. I imagine that this is supposed to be contrasted with just "standard" minimizing of a cross-entropy loss, in which case the logits in the toy example like this would diverge even when fixed all the remaining entries. Yet it is unclear how this property by itself is related with overfiting/overconfidence of a standard training method.

---

> ### Author Response · Authors · 2023-11-17
> **Author response**
>
> We would like to thank the reviewer for their thoughtful review, for agreeing that this is an important research question, and for supporting us in our work. We can see that you invested a lot of time to understand what is going on with OKO. Therefore, we have taken your feedback very seriously. Please find our answers to your questions below:
>
> 1.) The desirability of the phenomenon highlighted in Theorem 1 is certainly dependent on the application. It is most desirable when one would like to err towards uncertainty for regions of the input space that are sparsely sampled. This might be beneficial for a safety-critical setting where it would be better for the network to indicate less certainty in sample regions with little data rather than having the network assign high certainty from just a sample or two. We can include this point in an updated version of our manuscript, as well as Proposition 3, in the main text in the next draft.
> When one minimizes a function like $f(x,y) = -x-y+(-x+y)^{2}$, that has negative slope for $f(t,t)$, but is strictly convex with a unique minimizer  when $x$ or $y$ is fixed, gradient descent will typically follow a zig-zagging path that slows its divergence to $(\inf,\inf)$, unlike a function like $f’(x,y) = -x^{3}-y^{3}$ that will directly shoot of to $(\inf, \inf)$. We suspect this zig-zagging phenomenon helps keep the logits from diverging so quickly with OKO. We can include these example functions along with a plot of the gradient descent path to illustrate this. Again this isn’t a completely definitive characterization of OKO’s optimization path, but we feel this is still somewhat elucidating.
>
>
>
> 2.) We note that since both cross entropy $H(P, Q)$ and KL divergence are not symmetric, our definition of relative cross entropy is not the same as the reverse KL divergence, which is $H(Q, P) - H(Q)$. In fact, the RC can be negative and as we have shown in Lemma 2, the expected RC is 0 for calibrated classifiers. For Lemma 1, this is claiming that for data points where the predictor is wrong and overconfident, with the correct label being predicted with less than random chance, the RC value is positive and captures the overconfidence. We will clarify this and also strengthen the theorem.
>
>
> We hope that this addresses your concerns. Let us know if there’s anything else you’d like to know. We will update our manuscript to better clarify these points asap.

---

> ### Author Response · Authors · 2023-11-20
> **Propositions 2 and 3**
>
> We have added some discussion (after Proposition 2) and a figure (Figure 6 in the Appendix) to help elucidate Proposition 2 (see updated PDF of the manuscript). Please let us know if we can do more to help explain this result.
>
> Unfortunately, it is not feasible to move Proposition 3 to the main text. We are already very tight on space and the proposition and the contextualization that is necessary for it will require a rather large amount of space (alas, there is no extra page for rebuttal or camera-ready).

---

> > ### Comment · Reviewer_se1z · 2023-11-22
> >
> > Thank you. Yes, your example regarding the Proposition 3 is indeed enlightening, and you are of course right about the distinction between relative cross-entropy and the KL-divergence. It was a bit silly overlook on my part.
> >
> > I am still doubtful about whether the phenomenon highlighted in Proposition 1 is desirable. I certainly understand that it's undesirable for any learning algorithm to make high-confidence prediction from a sample or two. Yet, the theorem, as it is stated concerns low confidence prediction in a scenario where small _fraction_ of a training set is supported on what authors deemed as _low entropy regime_ -- but in some scenarios with large overall volumes of data, this might still make up millions of examples -- enough to potentially make much more confident prediction in similar situation in future.
> >
> > Nevertheless, the authors address a rather important question, demonstrate solid experimental evidence for efficacy of their method, and provide theoretical insight into some of its properties in toy scenarios. I raised my score to 8.

---

> > > ### Author Response · Authors · 2023-11-23
> > >
> > > Thank you for your response and for increasing your score! We appreciate your thorough feedback throughout the rebuttal. It has definitely helped us improve the clarity of our theorems. It is nice to see that there are reviewers who take the time to engage with the authors, just to better understand their work and help improve the manuscript. :)

---

### Author Response · Authors · 2023-11-17
**General response**

We would like to thank all reviewers for their time and effort. We feel that the reviews have been constructive and helpful in improving our manuscript. In our general response we’d like to respond to two issues that have been raised several times by different reviewers.

**Hard vs. Soft Loss**:

Some reviewers (in particular, reviewer ZKiJ, S9eW, and kv69)  were wondering about the (performance) differences between hard and soft OKO losses and why we think that the hard loss works better in practice. We have used this week to run additional ablation experiments that try to answer this question empirically and also thought about what may be happening here from a theoretical point of view. Empirically, we find that the soft loss optimization produces model predictions that tend to be more uncertain compared to the predictions obtained from using the hard loss. The soft loss optimization appears to result in output logits whose probability mass is spread uniformly across classes and, thus, produces probabilistic outputs with high entropy (often close to $\log{C}$). These uniformly spread out probabilistic outputs lead to worse ECEs, where differences between the hard and soft loss optimization are more substantial for MNIST and FashionMNIST than for CIFAR-10 and CIFAR-100 respectively.


The test set accuracy of models trained with the soft loss is substantially worse in all balanced class distribution training settings (the differences appear to be more pronounced for MNIST and CIFAR-100 than for FashionMNIST and CIFAR-10 respectively), but classification performance is only slightly worse compared to the hard loss in the long-tailed distribution settings (it's still worse but not as notable as in the class-balanced settings because, perhaps, in the long-tailed settings more uncertainty is needed than in the class-balanced settings).

We imagine that using the soft loss in combination with OKO mitigates the overconfidence problem in neural networks twice:


1.) The output logits are aggregated across all examples in a set and, thus, the loss is computed for a set of inputs rather than a single input (here, an input is an image). This step happens in both the hard and soft loss optimization.

2.) Probability mass in the target distribution is spread out across the different classes in a set and, hence, transforms the majority class prediction problem --- which is what the hard loss optimizes for --- into a proportion estimation problem, which may make the model unnecessarily underconfident about the correct class.


Since the output logits aggregation step is part of both the hard and the soft loss optimization, the hard loss may better strike the balance between mitigating the overconfidence problem and potentially shooting the optimization into local minima that amplify uncertainty where uncertainty/underconfidence is actually not desirable (which could be what is happening in the second step).


**Fitting Theory to the Empirical Results**:


Several reviewers were unhappy with parts of our theoretical analysis. OKO shows very impressive calibration performance in practice, and we tried to offer theories of why OKO works so well. Thanks to the feedback we received from you, we realize that we may have overstated the importance/gravity of our theoretical analysis. We will tone down our claims in this area to make this clear. We ask the reviewers to re-evaluate their criticism in the light that OKO’s empirical performance is what we want to communicate, and the theory is merely there to provide intuitions of why it might work so well.

Briefly, we believe that the standard cross-entropy (CE) loss encourages overfitting to its objective: it produces high confident estimates with low entropy in many of its predictions. We show that OKO provides some form of entropic regularization, and thereby calibration. We justify this entropic regularization framework by introducing a new form of entropic calibration measure. That measure captures the higher entropy levels of OKO (i.e. entropy is closely related to calibration). We show that for certain toy datasets, the entropy of the predictions are high (and higher than CE).

Again, we would like to emphasize that the theory isn’t meant to justify that OKO is a provably better approach than CE. It is mainly included to enlighten the expected benefits of the OKO loss, while the empirical results actually substantiate the fact that OKO performs much better than CE, as well as corroborate our theoretical intuitions.

We will update the PDF of the submission early next week to integrate all of your suggestions and fix any mistakes that have been pointed out. We would ask the reviewers to take the time to have a look. We are confident that the reviews have helped us improve our work, and we would please ask you to update your scores if your criticisms/concerns have been addressed :)

The Authors

---

> ### Author Response · Authors · 2023-11-20
> **Updated PDF of our manuscript (+ additional results and plots)**
>
> Dear Reviewers,
>
> as promised, we have just updated the PDF of our manuscript to incorporate your feedback. Specifically, we applied the following changes to our submission,
>
> - **Hard vs. soft loss optimization**: We ran additional ablation experiments to compare the *hard* and the *soft* loss optimizations against each other. We added plots, showing the results from these experiments, to a new section F.5 in the Appendix. We compare test set accuracy and calibration performance across all training settings (class-balanced and heavy-tailed) and for all datasets (our findings are outlined in the comment above). In addition, we added a subsection as part of this new section on "*Why soft loss optimization produces less accurate classifiers?*" to provide a well-informed intuition about why the soft loss produces worse classifiers than the hard loss. This subsection can be found in F.5.1.
>
> - **Better presentation of the theory**: We changed the wording in the abstract, updated the introduction/contributions, and the conclusion, to reflect the reviewers' criticisms about how some parts of the theory were presented, and applied the changes that we outlined above in **Fitting Theory to the Empirical Results** to our manuscript. In addition, to better clarify Theorem 1, we added a new paragraph to the main text (see Theorem 1 on page 5). To help elucidate Proposition 2 (as pointed out by Reviewer se1z), we added a figure demonstrating how a function that is convex in $x$ and $y$ meanders (see Figure 6 in the Appendix).
>
> - **Related works**: We added two sentences + references about *error-correcting output coding* (ECOC) to the related works section, as suggested by Reviewer S9eW.
>
> - **Typos / writing**: We fixed all typos (and ambiguity in the notation) that the reviewers pointed out.
>
> The Authors

---

### Meta-Review · Area_Chair_z2c1 · 2023-12-09

**Metareview:**

To alleviate poor calibration of some ML models, this paper proposes to generalize the entropic loss function to sets instead of individual examples. According to the authors, the main rationale is the calibration is inherently a properties of sets and not of instance-wise examples. Some theoretical properties of the new loss are investigated and strong numerical analysis illustrate that gain in both accuracy and calibration wrt previous methods.

Most of the reviewers agree that the contributions are significant and lean toward acceptance of the paper. However, the authors must update the current version to improve the writing. Below I noted some of them but reviewers have pointed to many others.


> Presentation/writing issue

Several presentation issues have been pointed by reviewers but the authors did not take it into account in the paper update. Here are few points.

- Why introduce the soft-loss and then discard it automatically without benchmarks (besides ablation)? What does one gain from it? What would be the behavior of the solution of the soft-loss in the example of theorem 1? Why not just move it to the appendix altogether?

- It is a bit uncomfortable to have a main section "Properties of OKO" to describe one single property "OKO is less certain in low-data regions". Plus this is not formally state in any result but in one single toy example that they call theorem.

- The introduced measure of calibration ie *relative Cross-Entropy* is not really compared with previous one. A comparison wrt ECE at least must be done. What do we gain with this new measure?
Plus, this is introduced in section 4 "properties of OKO" but this seems unrelated with the rest of the paper, not even discussed in the conclusion. The link is perhaps in the implicit regularization part vaguely mentioned in the introduction but I could not see clear description in the main text.

- The figures are definitely too small. This was mentioned by reviewers and the authors did *nothing* about it.

- Some of the motivations are still quite unclear. The authors state multiple times that classical ERM works with individual examples and then fail to capture complex correlation on sets of data. Ok, I could not really find concrete example to illustrate that clearly.


I believe that some serious reorganization of the paper is needed.

**Justification For Why Not Higher Score:**

The paper has several presentation issues and some times over claims. The presented theoretical results seems quite weak and does not explains the behavior of the proposed method.

**Justification For Why Not Lower Score:**

The paper provides several new insight on calibration both theoretical and experimental and is above acceptance threshold.

---

### Decision · Program_Chairs · 2024-01-16

Accept (poster)